# BioTrove: A Large Curated Image Dataset Enabling AI for Biodiversity

Chih-Hsuan Yang[1,*], Ben Feuer[2,*], Zaki Jubery[1], Zi K. Deng[3], Andre Nakkab[2], Md Zahid Hasan[1], Shivani Chiranjeevi[1], Kelly Marshall[2], Nirmal Baishnab[1], Asheesh K Singh[1], Arti Singh[1], Soumik Sarkar[1], Nirav Merchant[3], Chinmay Hegde[2], and Baskar Ganapathysubramanian[1]

[1]Iowa State University, Ames, IA 50011, USA
[2]New York University, New York, NY 10003, USA
[3]University of Arizona, Tucson, AZ 85721, USA
[*]Joint first authors
Correspondence: `chinmay.h@nyu.edu`, `baskarg@iastate.edu`.

## Abstract

We introduce BioTrove, the largest publicly accessible dataset designed to advance AI applications in biodiversity. Curated from the iNaturalist platform and vetted to include only research-grade data, BioTrove contains 161.9 million images, offering unprecedented scale and diversity from three primary kingdoms: *Animalia* ("animals"), *Fungi* ("fungi"), and *Plantae* ("plants"), spanning approximately 366.6K species. Each image is annotated with scientific names, taxonomic hierarchies, and common names, providing rich metadata to support accurate AI model development across diverse species and ecosystems.

We demonstrate the value of BioTrove by releasing a suite of CLIP models trained using a subset of 40 million captioned images, known as BioTrove-Train. This subset focuses on seven categories within the dataset that are underrepresented in standard image recognition models, selected for their critical role in biodiversity and agriculture: *Aves* ("birds"), *Arachnida* ("spiders/ticks/mites"), *Insecta* ("insects"), *Plantae* ("plants"), *Fungi* ("fungi"), *Mollusca* ("snails"), and *Reptilia* ("snakes/lizards"). To support rigorous assessment, we introduce several new benchmarks and report model accuracy for zero-shot learning across life stages, rare species, confounding species, and multiple taxonomic levels.

We anticipate that BioTrove will spur the development of AI models capable of supporting digital tools for pest control, crop monitoring, biodiversity assessment, and environmental conservation. These advancements are crucial for ensuring food security, preserving ecosystems, and mitigating the impacts of climate change. BioTrove is publicly available, easily accessible, and ready for immediate use.

## 1 Introduction

AI advances are poised to play a crucial role in biodiversity conservation, ecology management, and agriculture. Already, AI tools have been shown to enable automated species identification, monitoring of ecological changes, and optimization of crop management [36, 5]. However, standard AI approaches for biodiversity applications persistently face major challenges. Training datasets are labor-intensive and costly to create; they cover only a narrow set of

38th Conference on Neural Information Processing Systems (NeurIPS 2024) Track on Datasets and Benchmarks.

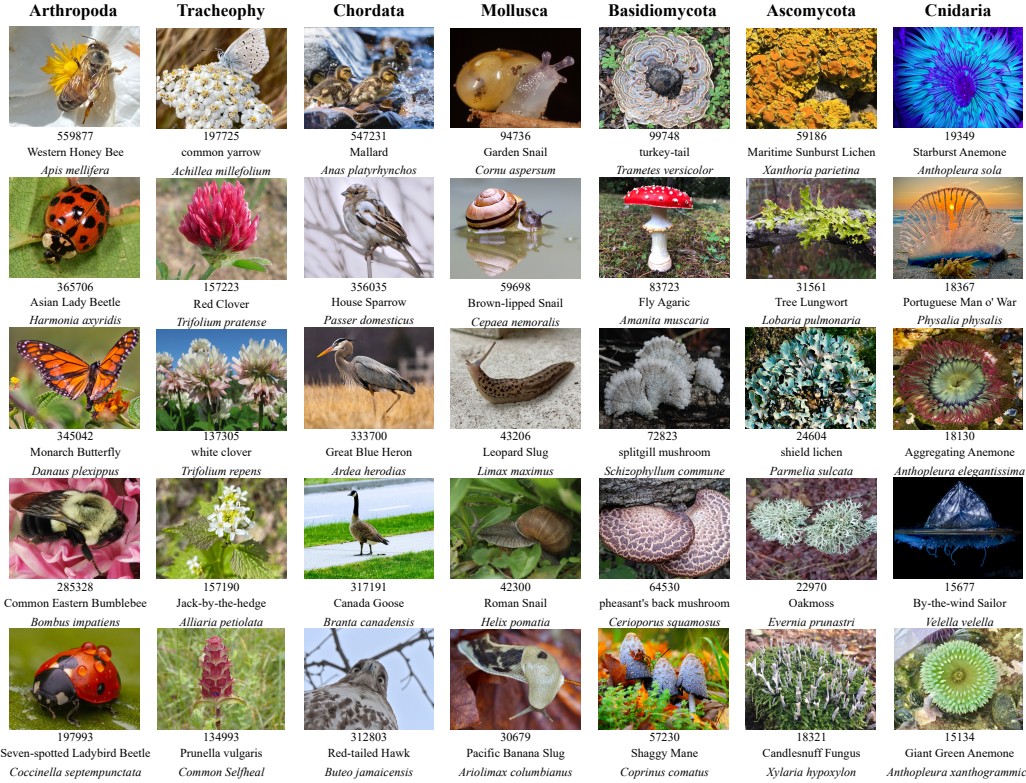

Figure 1: **Top Seven Phyla in the BioTrove Dataset.** *This figure displays the seven most frequently occurring phyla within BIOTROVE, which is curated to include data exclusively from the three primary kingdoms: Animalia, Plantae, and Fungi. For each phylum, the five most common species are shown, including their scientific names, common names, and the number of images per species. The phyla are ordered by species diversity, with the most diverse phylum on the right and the least diverse on the left.*

visual concepts; standard vision models excel at single tasks but require extensive retraining for new tasks; models often struggle with generalizing to unseen labels and new environments, limiting their effectiveness in real-world applications [34, 14]. Models that perform well on benchmarks often fail in the wild [12, 1]. Standard computer vision datasets (ImageNet and its successors) have significant limitations, including incorrectly labeled images, geographical and cultural biases, and overlapping or ill-defined labels, all of which impair the development of high-performant AI models [24]. Consequently, there is a critical need for large, diverse, accurately annotated datasets that are specific to biodiversity, ecology, and agricultural research [27, 23].

In response to this need, several datasets have been introduced. Perhaps the most well-known (raw) pool of biodiversity images on the Web is iNaturalist [42], from which several curated datasets have been sourced, among them being iNat2021 [41] with 2.7M images of over 10,000 species of plants, animals, and fungi. However, insects (which comprise a very large fraction of extant species) are under-represented in this dataset. IP102 [44], Insecta [10], and the more recent BIOSCAN-1M [13], are alternative datasets that focus on the Insecta Class. Perhaps the latest advance in such research is TREEOFLIFE-10M [39], which is currently the state-of-the-art dataset of text-annotated biological images, comprising 10M images with approximately 450K unique taxonomic classes.

In this paper, we contribute to advancing biodiversity AI research by curating and releasing **BioTrove**, a dataset comprising **161.9 million captioned images** across approximately **366.6K species**. This dataset surpasses all previous collections in both scale and diversity, representing the largest publicly available, "AI-ready" dataset of curated biodiversity images. Each image in BIOTROVE is paired with language data and spans a diverse range of

taxonomic groups, including *Reptilia* (reptiles), *Plantae* (plants), *Mollusca* (mollusks), *Mammalia* (mammals), *Insecta* (insects), *Fungi* (fungi), *Aves* (birds), *Arachnida* (arachnids), *Animalia* (animals), *Amphibia* (amphibians), and *Actinopterygii* (ray-finned fish). The dataset spans global regions, supporting robust training across diverse environmental contexts. Representative examples are shown in Figure 1, and additional details are provided on the project website.

Each image in BioTrove originates from the iNaturalist community science platform [42] and is annotated with detailed metadata, including the common name, scientific name, and complete taxonomic hierarchy. This curated metadata provides research-grade high-quality text annotations that enhance AI model training. Additionally, we open-source a data management pipeline, BioTrove-Process, to facilitate interaction with BioTrove metadata. With BioTrove-Process, researchers can efficiently filter and balance data by selecting specific taxonomic categories, adjusting for taxonomy level, and managing species distribution to reduce skewness. This enables users to create custom subsets that align with their research goals while maintaining consistency in species representation.

To showcase the capabilities of BioTrove, we introduce two technical contributions. First, we train and release BioTrove-CLIP, a suite of vision-language foundation models, using a subset, BioTrove-Train, consisting of approximately 40M images from BioTrove and representing around 33K species. This subset, constructed with BioTrove-Process, includes diverse taxa, specifically focusing on birds (*Aves*), spiders/ticks/mites (*Arachnida*), insects (*Insecta*), plants (*Plantae*), fungi (*Fungi*), snails (*Mollusca*), and snakes/lizards (*Reptilia*). These taxonomic classes were selected to capture a broad range of species—outside of charismatic megafauna—that critically impact biodiversity. The models exhibit robust generalization capabilities, demonstrating high zero-shot and few-shot performance on unseen taxa when using either common or scientific names. We anticipate that BioTrove-CLIP will serve as a valuable foundation for biodiversity-related applications and can be further fine-tuned for specific research needs.

Second, we rigorously quantify the performance of our foundation models on five existing fine-grained image classification benchmarks, as well as on three newly curated test datasets. We find that BioTrove-CLIP models comfortably achieve the state-of-the-art in certain settings, while both the original (OpenAI) CLIP model as well as BioCLIP [39] excel in certain other settings. We analyze these findings in further detail below, but overall we hope that our dataset can be used by the AI community as a testbed for further algorithmic and scaling research in fine-grained image recognition.

The remainder of this paper is organized as follows. Section 2 introduces the BioTrove dataset, the dataset's salient characteristics, and a comparison with previous work. Section 3 details our curation methodology. Section 4 introduces our newly proposed test datasets and their characteristics. Section 5 details our new BioTrove-CLIP models and their benchmark performance relative to previous work. Section 6 concludes with a discussion of limitations and potential future directions.

## 2    The BioTrove Dataset

**Characteristics.**    BioTrove comprises over 161.9 million images spanning 372,966 species. This dataset is an order of magnitude larger than existing biodiversity datasets, such as the state-of-the-art TreeOfLife-10M dataset, which it surpasses in *scale* by a factor of nearly 13.5× while maintaining comparable *species diversity*. Figure 1 shows representative image samples, while Figure 2 displays the distribution of samples across the seven major categories with the most frequently observed species. Additionally, Figure 3 illustrates the range of phyla, taxonomic classes, orders, and families represented in the dataset.

BioTrove includes only research-grade data and publicly accessible licensed content for research purposes from iNaturalist, which designates observations as research-grade once they meet strict validation criteria. To qualify, two or more experienced iNaturalist community members—naturalists, biologists, or citizen scientists—must agree on the species identification. Additionally, the observation must meet other requirements, such as a clear photograph and precise geolocation data. Recent experiments have shown that iNaturalist's Research Grade observations achieve approximately 97% accuracy, underscoring the reliability of this

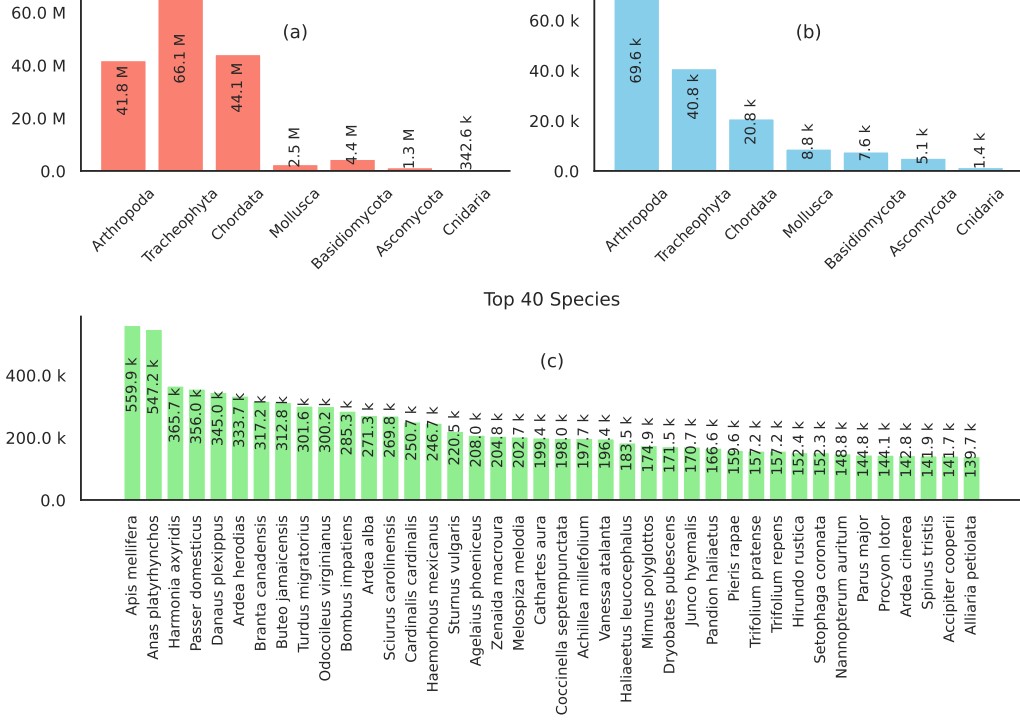

Figure 2: **Distribution of the BioTrove dataset.** *(a) Size of the top seven Phyla in the BioTrove dataset. (b) Species counts for the top seven Phyla. (c) The 40 highest occurring species in entire BioTrove dataset.*

community-based validation process [17]. Furthermore, iNaturalist continuously enhances data quality by refining validator criteria and implementing new data quality assessment measures, ensuring BIOTROVE remains a robust dataset for scientific use.

Each image sample in BIOTROVE is enriched with detailed, curated metadata that facilitates efficient filtering by species count and taxonomic information. The metadata includes common names, scientific names, and hierarchical taxonomic data, which enhances the usability of the dataset for AI model training. For the complete list of metadata fields, see Table 1.

Along with the dataset, we also release our data curation tooling pipeline: BIOTROVE-PROCESS, which enables users to easily access and manipulate the dataset. This pipeline allows researchers to select specific categories across different taxonomic levels, visualize data distributions, and effectively manage class imbalance according to their needs. It facilitates the downloading of specific images by their URLs and provides image-text pairs as well as user-defined chunks to support various AI applications. BIOTROVE-PROCESS thus enables users to define custom subsets of BIOTROVE with ease, making the dataset fully AI-ready and reducing barriers to follow-up research in biodiversity-focused AI.

**Dual-language text descriptions.** We adopt both common and scientific names since Latin is a low-resource language, and current AI models do not perform well on scientific names alone in a zero-shot manner. We found that a well-structured text description that integrates common names, scientific names, and detailed taxonomic hierarchies facilitates the learning of relationships between Latin and English terms, thereby improving the models' applicability in scientific contexts [6, 38, 43]. Moreover, incorporating the taxonomic hierarchy enables models to more effectively associate visual data with taxonomic terminology [25, 2]. This matches the guidelines suggested by BIOCLIP [39] to enhance model performance and generalization. **Privacy Measures:** The images of BIOTROVE were sourced from the

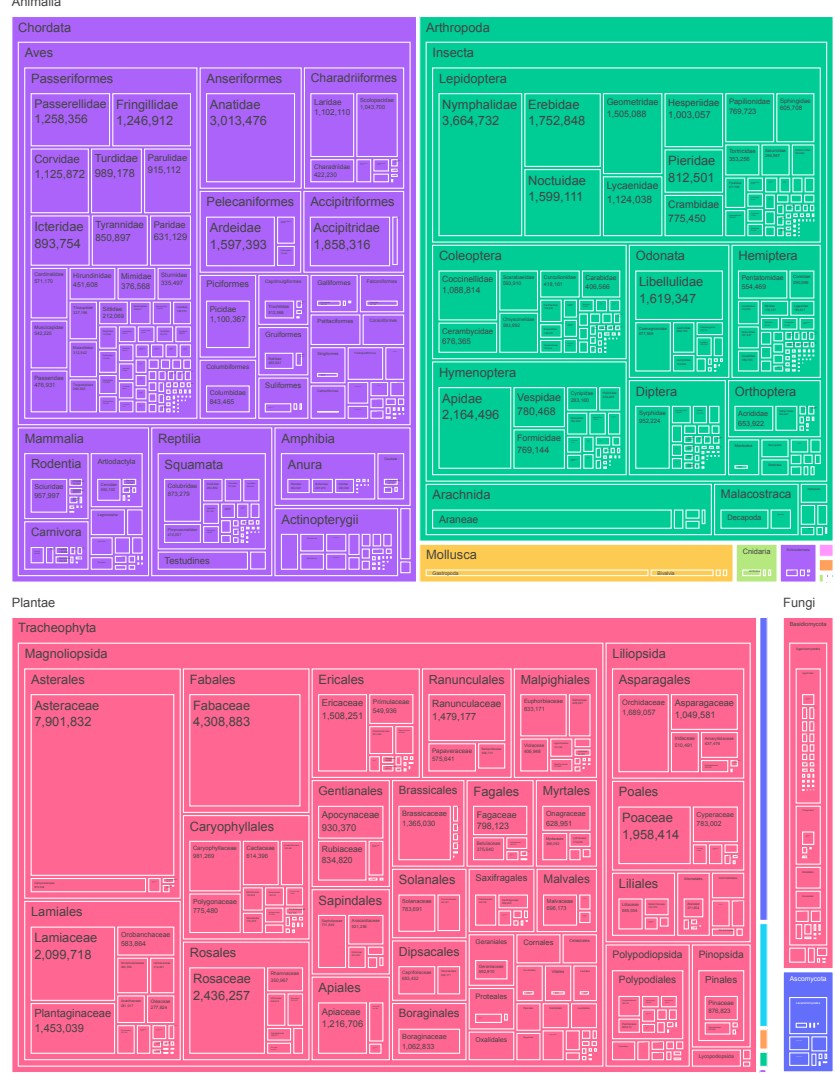

Figure 3: **Treemap diagram of the BioTrove dataset**, *starting from* Kingdom. *The nested boxes represent phyla, (taxonomic) classes, orders, and families. Box size represents the relative number of samples.*

Table 1: **Annotations provided in the BioTrove Dataset.**

| Text Type | Description |
| --- | --- |
| Common Name | Vernacular name (e.g., Western honey bee) |
| Scientific Name | Genus and species (e.g., *Apis mellifera*) |
| Taxonomic Name | Flattened taxonomy concatenated into a single string |
| Taxonomic Rank | Specific level in the hierarchy (e.g., subspecies, species) |

iNaturalist Open Dataset, whose metadata included Personally Identifiable Information (PII). This included information about observers, such as their usernames and sometimes their real names if they have chosen to share that information publicly. We removed all such fields to ensure that no PII is present in the metadata associated with BioTrove samples, ensuring the privacy of all contributors. **License:** During curation, we took care to include only images from iNaturalist Open Data, which are all licensed under either the `CC0`, or `CC-BY`, or `CC-BY-NC` licenses. This ensures that all our images are available for public research purposes. **Offensive Content:** Some of our URLs may point to images that users could find disturbing or harmful, such as photos of dead or dismembered animals. We retained these types of

images since they sometimes can provide valuable scientific data about wildlife, including information on predation events, roadkill, and other occurrences relevant to conservation and biodiversity studies. Although iNaturalist relies on user contributions and community moderation to maintain the quality and appropriateness of the data, we acknowledge that the vast and diverse nature of the data means that some offensive or inappropriate content might be present.

Our closest comparisons are with BioScan-1M (which appeared in NeurIPS 2023 Datasets and Benchmarks) and TreeOfLife-10M (which will appear in CVPR 2024). BioScan-1M focuses solely on the Insecta Class and provides scientific names, taxonomic ranks, as well as DNA barcodes. The TreeOfLife-10M dataset comprises 10.4 million images, integrating data from iNat2021 [41], BioScan-1M, and a fresh set of image samples sourced from the Encyclopedia of Life (EOL). It also supports dual-language labels and detailed taxonomic hierarchies and was used to train the BioCLIP vision-language model. See Table 2 for essential differences.

## 3   Data Collection and Curation Methodology

**Challenges with iNaturalist Open Data.**   All of BioTrove is sourced from the iNaturalist Open Data community science platform, which (in all) comprises over 280M biodiversity-relevant observations shared by users. However, there are still significant gaps in usability for AI research. The photos and metadata, although easily downloadable, are provided in four separate metadata sheets that are not ready to use. Taxa information is encoded as numerical IDs, requiring additional API calls and non-trivial lookups to convert these into common or scientific names. The multiple metadata sheets structure is fragmented across four separate files—photos, taxa, observations, and observers—adding complexity to data integration. Managing data balance and filtering out species with too few images can lead to biases toward common (charismatic) species and an imbalanced training process.

**Curation of BioTrove.**   The iNaturalist Open Dataset comprises a collection of 284.2 million images stored on an AWS S3 bucket as of 2024-09-27, with associated metadata provided across four separate CSV files (`photos`, `observations`, `taxa`, and `observers`). Details on each of these files are presented in Section A.5 in the Appendix. Although these files contain a wealth of valuable information, they are structured for rapid retrieval rather than AI-readiness. To address this, we curate the metadata into a streamlined, AI-optimized format.

We populate an SQL database with each CSV file as an individual SQL table, then create an aggregated table by joining `photos`, `observations`, and `taxa` on their relational columns, discarding irrelevant columns. In this aggregate table, we add a new column populated with

Table 2: ***Comparison*** *of BioTrove with existing biodiversity datasets.*

| Feature | BioTrove | TreeOfLife | BioScan |
|---|---|---|---|
| **Size** | 161.9 million images | 10.4 million images | 1.1 million images |
| **Diversity** | 366.6K species | 454.1K species | 8.3K |
| **Labels Provided** | Dual language (common and scientific names), detailed taxonomic hierarchies | Dual language (common and scientific names), detailed taxonomic hierarchies | Single language (scientific names), taxonomic ranks (family to species), DNA barcodes |
| **Data Source** | iNaturalist Open Dataset | iNaturalist, Encyclopedia of Life (EOL), BioScan-1M | Specimens from Malaise traps, DNA barcodes matched to BOLD |
| **Key Features** | Ready-to-use pipeline, reduce class imbalance, high-quality annotations, supports BioTrove-CLIP | Rich hierarchical representations, comprehensive metadata, supports BioCLIP | Focus on insects, high-resolution images, detailed taxonomic annotation, DNA codes |

the Amazon S3 URL and generate individual columns for taxonomic kingdom, phylum, class, order, family, genus, and species.

BioTrove includes only research-grade images from the *Animalia*, *Plantae*, and *Fungi* kingdoms, filtering out other domains to maintain a clear biodiversity focus. To achieve this filtering, we apply strict taxonomic criteria, ensuring only these three kingdoms are represented. The iNaturalist metadata files lack common names, so we reconstruct this information by cross-referencing species names from the iNaturalist Taxonomy DarwinCore Archive, updated monthly. This enriched metadata, including common names, is then appended to the SQL table. The final curated dataset is exported as parquet files, available for public access on HuggingFace.

**Data Filtering and Preprocessing.** As outlined, BioTrove includes structured metadata that is both comprehensive and easy to work with, featuring full taxonomic information and direct URLs to image files. To further support accessibility, we release an accompanying software pipeline that allows users to filter specific categories, visualize data distributions, and manage dataset imbalances effectively. These tools make it simple for researchers to interact with BioTrove, creating tailored subsets based on their specific needs. The iNaturalist data, sourced from citizen science contributions, has inherent variability in species representation, with some species documented extensively and others less so. To address this, our tools enable user-defined filters to exclude species with fewer than a set number of images and to cap image counts per species, thus supporting more balanced model training.

To further mitigate dataset imbalances (detailed in our experiments section), we use a semi-global shuffling strategy in which the data is organized into chunked tar files. These files are shuffled, divided into smaller groups, and then merged into larger batches to ensure a balanced species distribution within each batch. This method enhances dataset integrity, helping to prevent the overrepresentation of any single species across the batches.

## 4 Models and Benchmarks

We now showcase and demonstrate the utility of the BioTrove dataset by creating and benchmarking BioTroveCLIP, a new suite of vision-language foundation models for biodiversity.

### 4.1 BioTrove-Train

BioTrove-Train is a curated subset comprising approximately 40M samples and 33K species, focused specifically on seven taxonomic categories: *Aves*, *Arachnida*, *Insecta*, *Plantae*, *Fungi*, *Mollusca*, and *Reptilia*. As discussed previously, the BioTrove dataset is accompanied by a flexible pipeline that enables users to apply customized filtering to select specific categories or subsets based on research needs, thereby allowing researchers to generate their own training datasets. For BioTrove-Train, these seven categories were pre-selected due to their significant impact on biodiversity and agricultural ecosystems, as well as their relative underrepresentation in standard image recognition models. Unlike megafauna, which are typically well-represented in existing models, these categories address unique challenges in biodiversity-focused AI.

This subset comprises data posted on iNaturalist prior to 2024-01-27. We applied strict filtering criteria to ensure high-quality data, excluding species with fewer than 30 images and capping the maximum number of images per species at 50,000. To maintain balance, we employed a semi-global shuffling method, organizing the data into mini-batches of approximately 50,000 samples. From these, 95% were randomly selected for training and validation, while the remaining 5% were reserved for testing. Detailed statistics can be found in Table 3.

### 4.2 New Benchmarks

We created three new benchmark datasets, all of which are non-overlapping curated subsets of the BioTrove dataset. These benchmarks focus on fine-grained image classification within the seven taxonomic categories: *Aves*, *Arachnida*, *Insecta*, *Plantae*, *Fungi*, *Mollusca*, and *Reptilia*. All benchmarks presented here are independent and strictly within these seven categories, without overlapping with each other or with the BioTrove-Train subset.

Table 3: **Training data sources used in BioTrove-Train and Diversity in Different Taxonomy Levels**. *We integrate taxonomic labels into the images.*

| Dataset | Description | Images | Unique Classes | Level | Uniques |
|---|---|---|---|---|---|
| TreeOfLife-10M | Dataset combines a subset of iNaturalist, Encyclopedia of Life (EOL), BioScan-1M. | 10.4M | 454,103 | kingdom | 3 |
| | | | | phylum | 14 |
| | | | | class | 50 |
| | | | | order | 311 |
| | | | | family | 1692 |
| BioTrove-Train | One subset of BioTrove with size 40M. | 39.9M | 33,364 | genus | 11506 |
| | | | | species | 33364 |

Additionally, we report results on several established benchmarks from the literature (see Table 4).

**BioTrove-Balanced.** To ensure balanced species representation across the seven key taxonomic categories, we curate the BioTrove-Balanced benchmark. Each category includes up to 500 species, with 50 images per species, resulting in a total of 112,209 images. This balanced dataset provides a consistent foundation for model performance evaluations. The exact species counts for each category are detailed in Table 7 (see Appendix).

**BioTrove-Unseen.** To assess the ability of models to generalize to previously unseen species within the seven categories, we curated the BioTrove-Unseen benchmark. This dataset includes species from BioTrove-Train with fewer than 30 instances, ensuring they were unseen during training. Each species is represented by at least 10 images, with a total of 11,983 images. This benchmark tests the models' robustness on rare species not encountered during training.

**BioTrove-LifeStages.** The BioTrove-LifeStages benchmark evaluates the model's ability to recognize species across different developmental stages, focusing on insect species that exhibit significant visual variations throughout their life cycle. This dataset contains 20 labels representing four life stages (egg, larva, pupa, and adult) for five distinct insect species. The data was collected via the observation export feature on the iNaturalist platform between February 1, 2024, and May 20, 2024, ensuring no overlap with the training dataset. This benchmark allows for comprehensive evaluations of model performance across various life stages (see Figure 4).

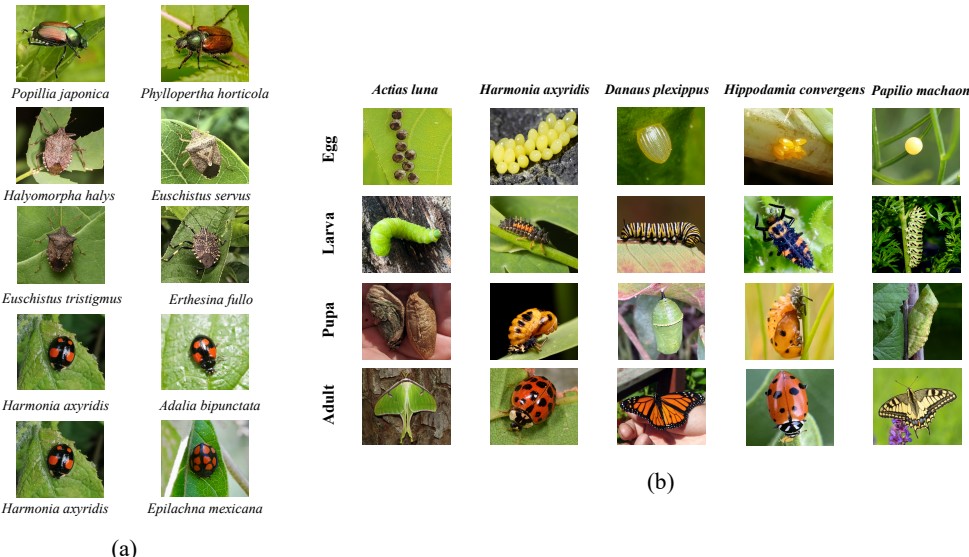

Figure 4: *(a) Example images from BioTrove-Unseen. (b) BioTrove-Life-Stages with 20 class labels: four life stages (egg, larva, pupa, and adult) for five distinct insect species.*

Table 4: Existing benchmark datasets; our novel datasets are described separately in section 4.2.

| | Name | Description | Examples | Classes | Labels |
|---|---|---|---|---|---|
| Anim | Birds 525 | Scraped dataset of bird images from web search [31]. | 89 885 | 525 | Taxonomic |
| | BioCLIP-Rare | Subset of species in the IUCN Red List categories: Near Threatened through Extinct in the Wild (iucnredlist.org). | 12 000 | 400 | Taxonomic |
| Plt & Fun | Fungi | Expert-labeled images of Danish fungi [30]. | 1000 | 25 | Scientific |
| | DeepWeeds | Weed images collected in situ from eight rangelands across northern Australia [29]. | 17 509 | 9 | Common |
| Inse | Confounding Species | Dataset evaluating models on challenging visually similar species pairs [4]. | 100 | 10 | Mixed |
| | Insects-2 | Mixed common and scientific name classification for insect pests [44]. | 4080 | 102 | Mixed |

## 4.3  BioTrove-CLIP: New vision-language foundation models for biodiversity

We use BioTrove-Train to train new CLIP-style foundation models and then evaluate them on zero-shot image classification tasks. Following the implementation of Stevens et al. [39], we utilize a ViT-B/16 architecture initialized from the OpenAI CLIP weights [33], and train for 40 epochs. We also train a ViT-L/14 model from the MetaCLIP [45] checkpoint for 12 epochs and a ViT-B/16 from the BioCLIP checkpoint for 8 epochs. All training hyperparameters are included in the Appendix (Section A.8). We compare with OpenAI's ViT-B/16 CLIP model, the BioCLIP ViT-B/16 checkpoint, and MetaCLIP-CC ViT-L/14. We publicly release all code needed to reproduce our results here.

## 5  Experimental Results

**Metrics.** We evaluate model performance using top-1 zero-shot accuracy across all benchmark datasets. For datasets containing taxonomic information, we report accuracy based on scientific names, ensuring fine-grained classification. For datasets that lack explicit taxonomic details, we use the category labels as defined by the original benchmark authors. We compute an aggregate performance metric, which represents the weighted average accuracy over all unique class labels across the benchmark suite. This aggregate metric provides an overall view of model performance across diverse tasks.

To account for statistical variability, we include 95% confidence intervals for all reported metrics, calculated using the binomial proportion confidence interval method (denoted by $\pm$). This provides a robust understanding of the performance and reliability of our results. As suggested during the review process, we incorporated this statistical analysis to strengthen the evaluation of our models.

**Overview of results.** In Table 5, we report the results of our core benchmark suite. At a high level, we observe that BioTrove-CLIP variants achieve the **best accuracy averaged over benchmarks**. In particular, they perform extremely well on BioTrove-Balanced (a remarkable **91.1** top-1 accuracy over 2250+ class labels). BioTrove-CLIP also does very well on the Fungi dataset (even though the Fungi class is not central to BioTrove-Train), and the DeepWeeds dataset. Therefore, BioTrove-CLIP exhibits strong generalization capabilities across diverse datasets.

We also observe that BioCLIP performs very well on BioTrove-Unseen and BioCLIP-Rare. The reasons might be that BioCLIP has seen approximately 450K species, and there might be nontrivial overlap with the species set in BioTrove-Unseen. On the other hand, it could be that BioTrove-CLIP suffers from forgetting issues while training on

Table 5: **BioTrove-CLIP performance on various benchmarks.** *The top three rows are pre-trained checkpoints: OpenAI-B refers to OpenAI's ViT-B-16 model, BioCLIP-B refers to the BioCLIP ViT-B-16 model, and MetaCLIP-L refers to the MetaCLIP-cc ViT-L-14 model. The bottom three rows are BioTrove-Clip models fine-tuned on different checkpoints: BT-Clip-O (from OpenAI-B), BT-Clip-B (from BioCLIP-B), and BT-Clip-M (from MetaCLIP-L). Benchmark abbreviations: BTU (Biotrove-Unseen, n=300), BTB (Biotrove-Balanced, n=2253), BCR (BioCLIP-Rare, n=400), F (Fungi, n=25), I2 (Insects-2, n=102), B (Birds-525, n=525), LS (Life-Stages, n=20), and DW (DeepWeeds, n=9). 95% confidence intervals (±) are included.*

| Model | BTU | BTB | BCR | F | I2 | B | LS | DW | Weighted Avg. |
|---|---|---|---|---|---|---|---|---|---|
| **OpenAI-B** | 12.9 ± 0.6 | 7.3 ± 0.15 | 10.9 ± 0.56 | 11.5 ± 1.98 | 10.2 ± 0.93 | 50.0 ± 0.33 | **56.5** ± 3.97 | 10.3 ± 0.45 | 14.7 |
| **BioCLIP-B** | **68.2** ± 0.83 | 62.2 ± 0.28 | **30.2** ± 0.82 | 45.1 ± 3.08 | **20.8** ± 1.25 | **68.7** ± 0.30 | 18.0 ± 3.07 | **19.9** ± 0.59 | 58.5 |
| **MetaCLIP-L** | 24.9 ± 0.77 | 15.4 ± 0.21 | 20.5 ± 0.72 | 24.6 ± 2.67 | 16.1 ± 1.13 | **70.1** ± 0.30 | **64.3** ± 3.83 | 14.7 ± 0.52 | 25.0 |
| **BT-CLIP-O** | 47.1 ± 0.89 | **91.1** ± 0.17 | 22.9 ± 0.75 | 43.2 ± 3.07 | 16.5 ± 1.14 | 47.8 ± 0.33 | 28.0 ± 3.59 | 17.0 ± 0.56 | **70.8** |
| **BT-CLIP-B** | **53.8** ± 0.89 | **82.2** ± 0.22 | **23.7** ± 0.76 | **53.9** ± 3.09 | **16.9** ± 1.15 | 57.1 ± 0.32 | 15.0 ± 2.86 | 18.4 ± 0.57 | 67.2 |
| **BT-CLIP-M** | 44.3 ± 0.89 | **91.1** ± 0.17 | 21.8 ± 0.74 | **54.7** ± 3.09 | 5.1 ± 0.68 | 42.5 ± 0.32 | 26.3 ± 3.52 | **49.9** ± 0.74 | **69.5** |

BioTrove-Train. For BioCLIP-Rare, the dataset is a subset from EOL which BioCLIP did not see before, but TreeofLife contains the majority of the EOL dataset.

**Limitations**. We also evaluated all models on the challenging Confounding-species benchmark introduced in [4], but find that all models perform at or below random chance and do not report results here; this could be an interesting avenue for follow-up work.

In Table 8 in the Appendix, we report model performance at different levels of the taxonomic hierarchy. Generally, we find that models trained on web-scraped data perform better with common names, whereas models trained on specialist datasets perform better when using scientific names. Additionally, models trained on web-scraped data excel at classifying at the highest taxonomic level (kingdom), while models begin to benefit from specialist datasets like BioTrove-Train and Tree-of-Life-10M at the lower taxonomic levels (order and species). From a practical standpoint, this is not problematic: BioTrove-CLIP is highly accurate at the species level, and higher-level taxa can be deterministically derived from lower ones.

Addressing these limitations will further enhance the applicability of models like BioTrove-CLIP in real-world biodiversity monitoring tasks.

## 6 Concluding Discussion

We introduce BioTrove, the largest publicly accessible dataset designed to advance AI for biodiversity applications. This dataset, curated from the iNaturalist community science platform, includes 161.9 million images, surpassing existing datasets in scale by an order of magnitude. We anticipate that BioTrove will enable the development of AI models that can enable various digital tools ranging from pest control strategies, crop monitoring, and worldwide biodiversity assessment and environmental conservation.

We also believe that BioTrove can be used as a unique testbed for measuring progress on fine-grained image recognition. The success of BioTrove-CLIP on BioTrove-Unseen underscores the importance of scaling up per-category sample size, or vertical scaling [9], in achieving high accuracy on long-tailed extreme-imbalance classification. However, BioCLIP continues to exhibit superior performance on several datasets, and we believe that this is because TreeofLife-10M contains an order-of-magnitude more *classes* (species) than BioTrove-Train. We invite the AI community to create new subsets of BioTrove with varying degrees of balance and species diversity and use our tooling to measure model performance against current benchmarks.

## Acknowledgements

We acknowledge support from the AI Research Institutes program supported by NSF and USDA-NIFA under AI Institute for Resilient Agriculture, Award No. 2021-67021-35329, and the NAIRR program for computing support.

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

# A Appendix

## A.1 Background on CLIP and zero-shot classification

Unlike traditional vision models, CLIP jointly trains an image encoder and a text encoder to predict the correct pairings of a batch of (image, text) examples, leveraging natural language supervision to enhance generalization [33]. CLIP's approach allows it to learn from a wide variety of images and their associated textual descriptions, making it more flexible and general compared to standard vision models. This flexibility is crucial for in various domains, including biodiversity monitoring and agriculture. For instance, CLIP models analyze digital plant specimen images, aiding in pre-processing and filtering for further analysis for agriculture purposes [19, 20]. As for biodiversity, WildCLIP and KI-CLIP facilitate wildlife observation and monitoring with high accuracy and effectiveness in data-sparse settings [11, 26]. These examples underscore the importance of developing and utilizing comprehensive datasets to fully leverage the capabilities of CLIP models in advancing biodiversity and agricultural research.

## A.2 The value of taxonomic information

Taxonomic classification, the hierarchical arrangement of organisms into categories based on shared characteristics, is foundational in biological sciences. Taxonomy underpins various scientific, ecological, and agricultural applications. It allows for precise identification and classification of species, which is fundamental for understanding biodiversity and monitoring ecosystems. For instance, accurate species identification can aid in tracking invasive species, as noted in studies such as [37]. In agriculture, detailed taxonomic information helps in identifying pests and beneficial species, thereby improving pest control strategies and crop management; supports ecological research by providing insights into species interactions, distribution patterns, and evolutionary relationships [13]; and is essential for policy-making and conservation planning [35].

## A.3 Scientific versus common names

Although we identify the importance and need to include taxonomic information in the dataset for biodiversity, one potential challenge is the fact that this information is mostly in Latin for which text embedding models often exhibit suboptimal performance due to its status as a low-resource language [40]. Nonetheless, Latin remains indispensable as it is the standard for representing scientific names and taxonomic classifications. We therefore integrate common names, scientific names, and detailed taxonomic hierarchies. We believe that such an "all-encompassing" approach facilitates the learning of relationships between Latin and English terms, thereby improving the models' applicability in scientific contexts [6, 38, 43]. Furthermore, incorporating taxonomic data into the training process significantly enhances the multimodal capabilities of the models, enabling them to associate visual data with taxonomic terminology [25, 2].

## A.4 iNaturalist, iNaturalist Open Data

iNaturalist is an online social network for sharing biodiversity information and learning about nature. It serves as a crowdsourced species identification system and organism occurrence recording tool. Users from around the world upload images, making the continuously updated dataset valuable for AI applications in biodiversity and research. Each photo includes detailed metadata: copyright status, location, uploader, time, and taxonomic classification. This diversity in image sources makes iNaturalist an excellent dataset for training AI models intended for real-world applications [41, 28, 7, 3]. Despite its vast and diverse data, iNaturalist is not directly optimized for AI researchers: arranging this data for use in AI models like CLIP is not straightforward. Each photo has its own page on the iNaturalist website, making it difficult to download images along with all the necessary information in a streamlined manner.

The iNaturalist Open Dataset aims to address some of these challenges. It is one of the world's largest public datasets of photos of living organisms, structured as a "bucket" of

images stored using Amazon Web Service's Simple Storage Service (S3). The dataset includes multiple resized versions of each photo, allowing users to download the size most useful to their research.

Additionally, the dataset provides four tab-separated CSV files representing observations, observers, photos, and `taxa_id`. These files are generated monthly, capturing a snapshot of the continually changing iNaturalist data. The images in the iNaturalist Open Dataset are licensed under either CC0, CC-BY, or CC-BY-NC and are open for public research. Photos with a CC0 license can be attributed as "[observer name or login], no rights reserved (CC0)". Photos with other Creative Commons licenses can be attributed as "© [observer name or login], some rights reserved ([license abbreviation])".

## A.5 iNaturalist Details

Each image in the iNaturalist Open Dataset can be associated with its appropriate metadata through a group of four metadata CSV files, representing photos, observations, taxa, and observers.

The photos metadata file contain nine distinct columns of metadata information of each photo. Of these columns, only photo_id and observation_uuid are relevant for us. The value of photo_id is a identifier number used to access individual photos, the photo's iNaturalist page can be found by constructing a URL in this format: https://www.inaturalist.org/photos/[photo_id]. The value of observation_uuid indicates which observation the photo is associated with, it is used to map the photos metadata to the observations metadata.

An observation represents one user submission of a species encounter to the iNaturalist website. One observation can have multiple photos of the same species but never multiple species. The observation metedata file contains eight distinct columns of metadata information on each observation. The columns relevant to us are observation_uuid, quality grade, and taxon_id. Each observation is given a unique number identifier indicated by its observation_uuid. iNaturalist has its own system to determining the quality of an observation and its associated photos, quality_grade represents this and can range from "Casual", "Research Grade", or "Needs ID". The value taxon_id indicates the species is represented in the observation, it is used to map the observations metadata to the taxa metadata.

The taxa metadata file contains information about each specific taxon in iNaturalist, it has has six distinct metadata columns. The columns relevant to us are taxon_id, name, ancestry, and active. Each specific taxon in iNaturalist has a unique identifier number associated with it, this is its taxon_id. This taxon_id will map to the scientific name of the taxon which is represented in the name metadata column. Each taxon also has associated with it a taxonomic ancestry, this is represented as a string of taxon_ids concatenated together with "\" like so "48460/1/47115/47584/1051154". The active column indicated whether the taxon is currently in use in iNaturalist.

The observer metadata file comtains information about each user within the iNaturalist site. For the purpose of machine learning research none of its three metadata columns are relevant.

While the iNaturalist Open Dataset metadata files provide a plethora of interesting information, its structure makes it inherently cumbersome to use for research. To solve this, we aggregate and process the iNaturalist metadata into a concise and streamlined format for easy query and usage.

First, the respective CSV files are used to populate a SQL database with each CSV file as its own SQL table. A new aggregate SQL table is created that joins the photos, observations, and taxa tables on its relational columns. Only the metadata columns we deemed relevant are kept and the extraneous non-useful metadata columns are discarded.

One of the difficulties working with the base iNaturalist metadata files is that it does not contain the image URL, information that is critical in image downloads. We include a new column in the aggregated metadata table that explicitly links to the Amazon S3 URL in which the image is hosted.

Table 6: Comparison of BioTrove with other biodiversity datasets.

| Dataset | BioTrove | Wildlife Insights | TreeOfLife | BioScan | iNaturalist 2017 [42] | iNaturalist 2019 [16, 15] | GBIF Backbone [8] |
|---|---|---|---|---|---|---|---|
| Size | 161.9 million images | 148.8 million images (52.6M wildlife images) | 10.4M images | 1.1M images | 675,170 images | 13.1M images | 7.5M records |
| Diversity | 366.6K species | 3,682 species | 454.1K species | 8.3K species | 5,089 species | 166.8K species | Millions of species |
| Labels Provided | Common/scientific names, taxonomic hierarchies | Species, location, timestamps, behavioral tags | Common/scientific names, taxonomic hierarchies | Scientific names, taxonomic ranks (family-species), DNA barcodes | Common/scientific names, taxonomic ranks (genus-species) | Common/scientific names, taxonomic ranks (genus-species) | Species names, OTU identifiers |
| Data Source | iNaturalist Open Dataset | Camera traps, sensors | iNaturalist, EOL, BioScan-1M | Malaise trap specimens, DNA-barcodes | iNaturalist | iNaturalist | Catalogue of Life, iBOL, UNITE, WoRMS, etc. |
| Key Features | AI-ready pipeline, high-quality annotations, supports BioTrove-CLIP | Automated processing, AI species recognition | Rich hierarchical data, metadata, supports BioCLIP | Insect-focused, high-resolution, taxonomic data, DNA codes | Imbalanced classes, fine-grained taxonomy | Large-scale species data, growth from 2017 | Comprehensive taxonomy, cross-referencing datasets |
| AI-Ready | Yes | Yes | Yes | Yes | No | No | No |

The BioTrove metadata file used for model training contains the metadata columns phylum, class, order, family, genus, species, scientific_name, common_name for the seven BioTrove categories *Aves, Arachnida, Insecta, Plantae, Fungi, Mollusca,* and *Reptilia.* To ensure that only images and metadata from the seven BioTrove categories appear in our final dataset we use the taxa table to find the taxon in our categories then use it in a SQL query on the ancestry column of our aggregated metadata table.

The taxonomic rank columns are also found utilizing the ancestry metadata column. A difficulty in working with the ancestry metadata is present in that there is not a clear indication of what taxonomic rank a taxon id represents the ancestry string. This problem is exacerbated due to the presence of taxonomic ranks and dsub ranks whose presence is variable across different species. As such, a custom function is applied to each row to dynamically find the rank of each taxon id in the ancestry and then appropriately populate the taxon id to a metadata column of that rank. This process results in all taxonomies rank represented as metadata columns; only phyllum, class, order, family, genus and species are kept in the BioTrove metadata file.

The scientific name of a species is found using the name metadata column of our aggregated metadata table. The common name of a species is also useful metadata information. Unfortunately, the iNaturalist Open Data metadata files do not contain the common name information of a species. To address this, we curate a lookup table of the common names in our dataset. This is obtained from the iNaturalist Taxonomy DarwinCore Archive, Having obtained the common names for each species, we append it to the BioTrove-specific metadata.

## A.6 Composition of BioTrove and Related Datasets

In Table 6, we compare BioTrove with existing large-scale biodiversity datasets. BioTrove comprises 161.9 million research-grade images, representing approximately 372,966 species, and significantly surpasses other datasets in terms of both diversity and scale.

## A.7 Composition of BioTrove-Train

See Figure 5 and Table 7.

## A.8 BioTrove-CLIP training details

We use BioTrove-Train to train new CLIP-style foundation models, and then evaluate them on zero-shot image classification tasks. Following the implementation of Stevens et al. [39], we utilize a ViT-B/16 architecture initialized from the OpenAI pretrained weights for our main model, and train for 40 epochs.

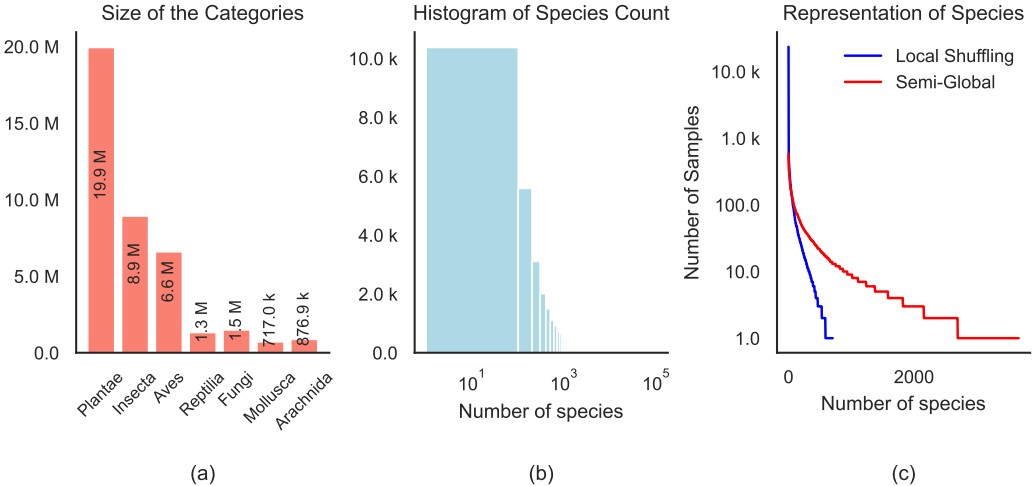

Figure 5: *BioTrove-Train Dataset Analysis: a) Consistent category distribution across BioTrove-Train and BioTrove-116M datasets. b) Species exhibit a long-tailed distribution. c) Impact of local vs. semi-global shuffling on species representation within training minibatches.*

Table 7: *Number of Unique Species in Each Category in BioTrove-Balanced.*

| Category | Number of Unique Species |
|---|---|
| Kingdom: Fungi | 281 |
| Kingdom: Plantae | 500 |
| Phylum: Mollusca | 147 |
| Class: Insecta | 500 |
| Class: Arachnida | 136 |
| Class: Reptilia | 189 |
| Class: Aves | 500 |

In addition, we also train a ViT-L/14 model from the MetaCLIP [45] checkpoint for 12 epochs, and a ViT-B/16 from the BioCLIP checkpoint for 8 epochs. We select the AdamW optimizer from Loshchilov and Hutter [22] along with a cosine learning rate scheduler, as this has previously been shown to perform well for CLIP pretraining [32]. We conduct twenty rounds of hyperparameter optimization using Ray Tune [21] to determine the optimal learning rate, $\beta_1$, $\beta_2$ and weight decay settings.

We train our models for a combined 10 days on 8xH100 nodes in bfloat16 precision [18] with gradient checkpointing, computing loss with local features, and utilizing static graph optimization for DDP.

## A.9 Additional BioTrove-CLIP results

In Table 8, we report model performance at different levels of the taxonomic hierarchy. Generally, we find that models trained on web-scraped data perform better with common names, whereas models trained on specialist datasets perform better when using scientific names. Additionally, models trained on web-scraped data excel at classifying at the highest taxonomic level (kingdom), while models begin to benefit from specialist datasets like BioTrove-Train and Tree-of-Life-10M at the lower taxonomic levels (order and species).

However, BIOTROVE-CLIP shows a performance decline at taxonomic levels below the species level. This is likely because our training metadata structure allows for classifications solely by referring to species information. From a practical standpoint, this is not problematic for the species in our test set since BIOTROVE-CLIP is highly accurate at the species level, and higher-level taxa can be deterministically derived from the lower ones.

Furthermore, the OpenCLIP and MetaCLIP baselines outperform BioTrove-CLIP on the life stages benchmark. This highlights the importance of retaining the general linguistic capabilities of the pretrained CLIP models for hybrid tasks.

Table 8: Performance Comparison Across Benchmarks: This table compares the performance of BC-iNat21 (trained solely on the iNaturalist 2021 dataset) and BT-Clip (trained from the BioCLIP checkpoint, originally trained on the TreeOfLife dataset). Metrics include Top-1 Accuracy and Top-5 Accuracy.

| Benchmark | BC-iNat21 Top-1 | BC-iNat21 Top-5 | BT-Clip Top-1 | BT-Clip Top-5 |
|---|---|---|---|---|
| BioTrove Unseen | 0.2100 | 0.3470 | 0.5380 | 0.8220 |
| *Fungi* | 0.4420 | 0.7550 | 0.5390 | 0.7590 |
| Life-Stages | 0.2867 | 0.8617 | 0.1500 | 0.8600 |
| DeepWeeds | 0.2057 | 0.6897 | 0.1840 | 0.5740 |
| *Insects-2* | 0.0103 | 0.0483 | 0.1690 | 0.5710 |
| *Birds-525* | 0.5030 | 0.6330 | 0.5710 | 0.7540 |
| BioCLIP-Rare | 0.1490 | 0.2790 | 0.2370 | 0.7600 |
| BioTrove Balanced | 0.5020 | 0.6450 | 0.5180 | 0.6610 |

### A.10 Additional BioTrove-CLIP Comparative Analysis

We conducted a comparative evaluation of the top-1 and top-5 zero-shot accuracy of the BioCLIP model, which was trained exclusively on the iNaturalist 2021 (iNat21) dataset, and the BioTrove-CLIP model, initialized from BioCLIP checkpoints originally trained on the TreeOfLife dataset. The comparison highlights the performance differences across various benchmarks, as presented in Table 9.

Our analysis shows that models trained on the BioTrove dataset consistently outperform those trained solely on iNat21, particularly in benchmarks such as BioTrove-Unseen, *Fungi*, and *Insects-2*. While certain benchmarks like Life-Stages and DeepWeeds show moderate differences, the results emphasize the advantages of training on BioTrove, leading to enhanced model accuracy and robustness.

The following table provides detailed performance metrics for both models across various benchmarks, comparing their top-1 and top-5 accuracy scores with associated confidence intervals.

Table 9: Performance Comparison Across Benchmarks: This table compares the performance of BC-iNat21 (trained solely on the iNaturalist 2021 dataset) and BT-Clip (trained from the BioCLIP checkpoint, originally trained on the TreeOfLife dataset). Metrics include Top-1 Accuracy and Top-5 Accuracy. BC-iNat21 refers to BioCLIP (iNat21), and BT-Clip refers to BioTrove-CLIP (BioCLIP checkpoint from TreeOfLife).

| Benchmark | BC-iNat21 Top-1 Acc. | BC-iNat21 Top-5 Acc. | BT-Clip Top-1 Acc. | BT-Clip Top-5 Acc. |
|---|---|---|---|---|
| BioTrove-Unseen | 0.2100 | 0.3470 | 0.5380 | 0.8220 |
| *Fungi* | 0.4420 | 0.7550 | 0.5390 | 0.7590 |
| Life-Stages | 0.2867 | 0.8617 | 0.1500 | 0.8600 |
| DeepWeeds | 0.2057 | 0.6897 | 0.1840 | 0.5740 |
| *Insects-2* | 0.0103 | 0.0483 | 0.1690 | 0.5710 |
| *Birds-525* | 0.5030 | 0.6330 | 0.5710 | 0.7540 |
| BioCLIP-Rare | 0.1490 | 0.2790 | 0.2370 | 0.7600 |
| BioTrove Balanced | 0.5020 | 0.6450 | 0.5180 | 0.6610 |

As demonstrated, the model trained on BioTrove exhibits superior performance in most categories, particularly when evaluated on rare and unseen species, underscoring the importance of diverse and large-scale datasets like BioTrove for enhancing biodiversity AI models.

