# Arboretum: A Large Multimodal Dataset Enabling AI for Biodiversity (*Supplemental Material*)

Chih-Hsuan Yang[1,*], Ben Feuer[2,*], Zaki Jubery[1], Zi K. Deng[3], Andre Nakkab[2], Md Zahid Hasan[1], Shivani Chiranjeevi[1], Kelly Marshall[2], Nirmal Baishnab[1], Asheesh K Singh[1], Arti Singh[1], Soumik Sarkar[1], Nirav Merchant[3], Chinmay Hegde[2], and Baskar Ganapathysubramanian[1]

[1]Iowa State University, Ames, IA 50011, USA
[2]New York University, New York, NY 10003, USA
[3]University of Arizona, Tucson, AZ 85721, USA
[*]Joint first author
Correspondence: `chinmay.h@nyu.edu`, `baskarg@iastate.edu`.

## 1  Dataset Documentation and Intended Uses

Arboretum is a 134.6M sample dataset designed to advance AI for biodiversity applications by providing a large-scale, accurately annotated multimodal dataset that includes images and corresponding textual descriptions for a diverse set of species. Arboretum aims to facilitate the development of AI models for species identification, ecological monitoring, and agricultural research. Additionally, we introduce three new benchmark datasets: Arboretum-Unseen, Arboretum-LifeStages, and Arboretum-Balanced.

## 2  URLs for Dataset Access

- **Project Website:** https://baskargroup.github.io/Arboretum/

- **Hosting Dataset:** https://huggingface.co/datasets/ChihHsuan-Yang/Arboretum

- **Package:** https://pypi.org/project/arbor-process/

- **Croissant Metadata Record:** The dataset and detailed information about the dataset can be found on Hugging Face: https://huggingface.co/datasets/ChihHsuan-Yang/Arboretum

## 3  Author Responsibility Statement

As the authors of this submission, we affirm that we bear all responsibility in case of any rights violations or ethical issues associated with this work. We confirm that the submitted work is original, and if it includes third-party content, it is used with proper permissions and attributions.

## 4  Hosting, Licensing, and Maintenance Plan

The dataset is hosted on Hugging Face. We ensured that only images licensed under `CC0`, `CC-BY`, or `CC-BY-NC` from iNaturalist Open Data were included, making them available for public research purposes.

## 5  Dataset Format and Accessibility

The metadata is provided in parquet format, which includes information such as photo ID, scientific name, detailed taxa information, common name, and photo URL. The Arbor-preprocess package facilitates easy access and processing of this data.

Preprint. Under review.

## 6 Long-term Preservation

Our dataset will be available for as long as the iNaturalist Open Dataset is maintained. Our pipeline and metadata will continue to function, ensuring long-term accessibility.

## 7 Structured Metadata

The metadata is provided in parquet file format and includes information such as photo ID, scientific name, detailed taxa information, common name, and photo URL.

## 8 Persistent Identifier

- **GitHub Repository:** https://github.com/baskargroup/Arboretum
- **Package:** https://pypi.org/project/arbor-process/