# OpenReview forum: "BioTrove: A Large Curated Image Dataset Enabling AI for Biodiversity"
_NeurIPS.cc/2024/Datasets_and_Benchmarks_Track — NeurIPS 2024 Track Datasets and Benchmarks Spotlight_

### Official Review · Reviewer_7g25 · 2024-07-24
**A carefully constructed and important dataset**

**Rating:** 7
**Confidence:** 3

**Review:**

# Quality
The dataset, originally sourced from iNaturalist, is very large and has been curated with domain expert input. It covers a diverse range of species across seven different taxonomic classes. The breadth and apparent quality of the dataset make it stand out as a potential new standard.

The experimental benchmarks, while limited in scope to smaller subsets of the data, appear to be well constructed and useful both as baselines and as starting points for further research.

# Originality
The concept of a dataset like this is not new, however the scale of the data is what makes this contribution significant.

# Significance
I believe the availability of a dataset of this magnitude and quality marks a significant step forward in the field.


# Pros
- Good quality wide scope dataset
- Apparently well-curated

# Cons
- No mammals in the dataset. This is a deliberate choice by the authors, but it would be nice to have the option of including them.
- Nit: Naming - why ARBORETUM? On first look it sounds like it would be a dataset of trees.

**Strengths:**

- Significant dataset due to scale and breadth of subject matter
- Well constructed
- Relevant to a number of domains, including potential interest beyond the original intended domain of biodiversity due to the scale of the dataset
- Three well-defined benchmark problems - 'balanced', 'unseen' and 'lifestages'

**Additional Feedback:**

No other feedback not captured above.

**Clarity:**

The paper is clear and well written. I have no concerns here. There are no obvious typos.

Nit: Line 223: the 'EOL' dataset is mentioned without any clarification of what it is.

**Correctness:**

Dataset construction appears sound and well documented.

Experiments on subsets of the data appear to be well constructed. Little space in the paper is given over to these experiments, but source code is provided and further details are provided in the appendix.

Code is provided to access and process the dataset, and to reproduce the experimental results.

**Documentation:**

Licence: In the paper, the authors note that the dataset was constructed from iNaturalist images with CC-0 CC-BY and CC-BY-NC licences. The github site states that the data on that site is CC-BY licensed. It would be useful to clarify in the README that the linked images in the collection are under a range of licences which include noncommercial.

Geographical distribution of images: The authors don’t address the geographical distribution of the images in the dataset. While this is not central to the dataset’s usefulness, it would be interesting to understand the rough distribution of where the images were taken (or the related question of the native habitat of the species), as this may be a source of hidden bias in the data.

It would be nice for the dataset to have a DOI. Hosting platforms such as Zenodo allow for the generation of a DOI, and dataset versioning.

The source code in the liniked github repository appears to cover the training / fine tuning of models and the experiments. However the tooling used to generate the dataset in the first place from iNaturalist is not obviously available (or at least not linked in the documentation). The authors should consider adding any code used for this process, or make it clearer where it is.

**Ethics:**

The dataset consists of links to user-generated images from iNaturalist. The authors note that all images selected are made available under creative commons licences. The authors note that there may be offensive content in the linked images including dead and dismembered animals, but that these are left in the dataset as there may be a legitimate scientific interest in such images. The authors note that they scrubbed any metadata fields which could contain PII about the original observers.  Since all images have explicitly been released under CC licences by their original authors, metadata has been scrubbed of PII and and the paper’s authors acknowledge and defend the potential for limited offensive content in the dataset I don’t see any significant ethical issues.

**Limitations:**

- Various limitations have been identified by the authors. Beyond the potential for providing additional tools and versioning capabilities to allow expansion of the dataset as noted above, I have nothing else to add.

**Opportunities For Improvement:**

- Further expansion of dataset (or make tooling available for others to do so easily). On a related note, a plan for future curation of the dataset, including the ability to publish errata and for others to build on the expansion of the iNaturalist corpus would be a welcome addition.

**Relation To Prior Work:**

Prior work is covered in the introduction in lines 37-44, and elsewhere in the introduction. While there is no explicit prior work section I believe there is enough information presented to place the work in context.

**Summary And Contributions:**

The authors present a curated dataset of around 130M images of over 320K different species of birds, plants, insects, reptiles, molluscs, arachnids and fungi. The images are drawn from the iNaturalist community science platform and annotated with scientific and common names and taxonomic information using expert input.

The scale of this dataset exceeds the current state of the art dataset, TreeOfLife-10M by an order of magnitude in number of images, while having a roughly comparable level of species diversity.

A number of subsets of the dataset are presented as being useful for specific tasks.

A subset of the dataset is used to train a set of CLIP models to demonstrate its utility, and the authors present a set of benchmarks to measure the effectiveness of the dataset for zero-shot and few-shot learning and other applications of interest.

---

> ### Author Rebuttal · Authors · 2024-08-17
>
> Thank you for your thoughtful comments and suggestions. We appreciate the opportunity to clarify our choices and future plans for the Arboretum dataset.
>
>
> **Regarding the name:** The name "Arboretum" was chosen to reflect both our team’s interests in agriculture as well as the symbolic "tree of life", which represents the interconnectedness of all living organisms and the cycle of life and death. This concept reflects our work towards establishing a comprehensive taxonomic hierarchy, as well as new evaluation datasets that span multiple stages in the organism’s life. This is why our benchmark suite includes ARBORETUM-LIFE-STAGES; this was specifically constructed to measure  a model’s deeper understanding of biodiversity. We note that this name is beginning to gain some visibility within the AI-for-agriculture community.
>
>
> **Regarding the exclusion of mammals:** This decision was somewhat deliberate. We were initially focused on AI applications directly related to agriculture where mammals are less relevant compared to insects and weeds. Moreover, as we discuss in Page 4 and the footnote on Page 6, larger mammals (particularly, charismatic megafauna) are already well-represented in popular image datasets, and identifying/tracking them is an established AI/ML task; the main challenges in image recognition arise from species such as plants, weeds, and insects which exhibit a remarkable amount of nuanced visual diversity across their lifecycle that is not easily captured by ML models. However, we do acknowledge the value of expanding the dataset to include mammals and are currently working to ensure future releases of Arboretum will include them. We believe this can enhance the dataset’s utility for broader biodiversity studies.
>
>
> **Regarding the future of Arboretum:** We are committed to its continuous development and expansion. We plan to explore different subsets or potentially the entire Arboretum dataset to fully understand its capabilities and maximize its potential. Including more species, as suggested, is an excellent idea that aligns with our goals. Additionally, we intend to develop tools that will enable others to contribute to and expand upon our work. We recognize the importance of providing mechanisms for publishing errata and facilitating continuous updates to ensure that Arboretum remains a reliable and dynamic resource.
>
> Overall, we are excited about the potential of Arboretum, and are committed to its ongoing improvement. We believe these planned enhancements will further strengthen its value to the research community. Thanks again for your valuable comments.

---

> > ### Comment · Reviewer_7g25 · 2024-08-30
> >
> > Thank you for all of your carefully-considered responses. I think the name change in particular is important as it makes the dataset more discoverable.

---

### Official Review · Reviewer_KAB1 · 2024-07-25
**Self-contained dataset, or repository of metadata?**

**Rating:** 3
**Confidence:** 5
**Correctness:** Yes
**Clarity:** Yes

**Review:**

The paper's claim of introducing the "largest public, 'AI-ready' dataset of curated biodiversity images" could be misleading. While Arboretum provides a valuable resource of 134.6 million image references and associated metadata, it doesn't appear to be hosting the images themselves. The actual image data remains on the iNaturalist servers, and Arboretum merely provides URLs to these images. This distinction is crucial, as it impacts the accessibility and usability of the dataset. Researchers still rely on iNaturalist's infrastructure to access the images, making Arboretum more of a metadata repository than a self-contained dataset. Therefore, it is difficult to consider this paper as an independent dataset paper. The paper should emphasize this distinction to avoid misrepresenting the nature of their contribution.

**Strengths:**

The paper improved upon the metadata of iNaturalist Open Data, a community science platform with over 190 million biodiversity-relevant observations. However, the raw data from iNaturalist was not readily usable for AI research due to several issues. For example, the photos and metadata were fragmented across four separate files, making data integration complex; taxa information was encoded as numerical IDs, requiring additional API calls and lookups to convert them into common or scientific names; there was a wide variation in the number of observations per species, leading to a severe imbalanced data issue.

The paper provides some benchmarks for the curated data.

**Additional Feedback:**

1. Provide a standalone data repository for the dataset.

2. Provide more benchmarks: e.g., baseline architectures such as ResNet; compact networks such as MobileNet.

3. Offer comprehensive comparisons with more previous datasets, such as iNaturalist 2017 to 2021.

4. Change the dataset name: Arboretum means cultivated botanical garden, which is different from what this dataset is about. Suggestion: Biosphere, since it claims to be the biggest biodiversity data. Or Compendium of Nature, since it provides well-curated summary metadata for existing snapshots of nature.

5. Provide more information about the iNaturalist dataset --- for example, biased nature of iNaturalist dataset that relies on citizen science. Image resolution, and other image statistics. The quality of iNaturalist dataset.

**Documentation:**

The [submission guide](https://neurips.cc/Conferences/2023/CallForDatasetsBenchmarks) states the recommended documentation frameworks, which the paper lacks:

> Dataset documentation and intended uses. Recommended documentation frameworks include datasheets for datasets, dataset nutrition labels, data statements for NLP, data cards, and accountability frameworks.

**Limitations:**

1. Limited Taxonomic Scope: The dataset focuses on only seven taxonomic classes, potentially limiting its applicability for broader biodiversity research. This could introduce biases in AI models trained on this dataset, making them less effective for tasks involving other taxonomic groups.


2. Potential Biases in iNaturalist Data: The data is sourced from iNaturalist, a citizen science platform, which may contain biases in terms of geographic location, species representation, and image quality. These biases could affect the performance and generalizability of AI models trained on Arboretum.


3. Limited Evaluation on Challenging Tasks: The paper reports limited evaluation on challenging tasks like the Confounding-Species benchmark, where all models performed poorly. This suggests that there is still room for improvement in developing AI models that can handle fine-grained distinctions between visually similar species.


4. Lack of Long-Term Monitoring Data: The dataset is a snapshot of iNaturalist data, lacking long-term monitoring data that could be valuable for studying ecological changes and trends over time. This limits the dataset's utility for certain research questions related to biodiversity conservation and environmental monitoring.

**Opportunities For Improvement:**

1. Provide a standalone data repository for the dataset.

2. Provide more benchmarks: e.g., baseline architectures such as ResNet; compact networks such as MobileNet.

3. Offer comprehensive comparisons with more previous datasets, such as iNaturalist 2017 to 2021.

4. Change the dataset name: Arboretum means cultivated botanical garden, which is different from what this dataset is about. Suggestion: Biosphere, since it claims to be the biggest biodiversity data. Or Compendium of Nature, since it provides well-curated summary metadata for existing snapshots of nature.

5. Provide more information about the iNaturalist dataset --- for example, biased nature of iNaturalist dataset that relies on citizen science. Image resolution, and other image statistics. The quality of iNaturalist dataset.

**Relation To Prior Work:**

Can be improved. For example, discussion of how the dataset differs from previous curation of iNaturalist dataset is not sufficient.

**Summary And Contributions:**

This paper introduces a new dataset for advancing AI in biodiversity applications. The dataset contains the metadata of 134.6 million images of 326,888 species, which the authors claim is the biggest dataset for biodiversity. The dataset covers seven taxonomic classes: Aves (birds), Arachnida (spiders/ticks/mites), Insecta (insects), Plantae (plants), Fungi (fungus/mushrooms), Mollusca (snails), and Reptilia (snakes/lizards). Each image URL is linked with common and scientific names, taxonomic details, and other metadata. The authors curated a subset of iNaturalist Open Data by consolidating fragmented metadata, reconstructing common names, and addressing data imbalance issues through filtering and shuffling. The paper is not a dataset in itself but a metadata repository that provides 134.6 million image references with curated metadata, while the actual images are hosted on iNaturalist servers. This reliance on external infrastructure for image access impacts the dataset's accessibility and usability, making it more of a metadata contribution than a standalone dataset. Sourcing the images to an independent data storage service such as [Amazon open data](https://aws.amazon.com/opendata/) would strengthen the value of the paper. It provides some benchmark, but for only one model. More benchmarks can improve the paper. Also, it would benefit from more comprehensive comparison of existing datasets, such as iNaturalist 2017 to 2021. Finally, Arboretum sounds pretty misleading, so it needs to be changed to another name that represents the dataset better.

---

> ### Author Rebuttal · Authors · 2024-08-17
>
> Thank you for your detailed and thoughtful feedback on our submission. We appreciate the opportunity to address the concerns raised and provide additional clarity.
>
> **Regarding the value of Arboretum as a stand-alone dataset:** we would like to clarify that while Arboretum indeed sources images from the iNaturalist open data, our contribution goes beyond merely providing metadata. We have significantly enhanced the dataset by adding ten new text descriptions for each image, which greatly improves its usability for AI and machine learning applications. This addition ensures that Arboretum is a rich, context-specific dataset that is both new and unique.
>
> **Regarding the matter of hosting the dataset ourselves:** we request the reviewer to consider the scale and nature of the Arboretum dataset. With nearly 135 million high-quality captioned images, it stands among the largest multimodal datasets available to the research community. Hosting this dataset on our own would involve considerable financial (and environmental) burden, especially the source (iNaturalist) dataset is maintained and updated by a dedicated staff.  The iNaturalist Open Dataset, in particular, is updated monthly, making it one of the most reliable public datasets of living organisms' photos. Managing a concurrent version of such a dataset on our own would likely lead to redundancy and might not efficiently reflect the most current data.
>
> We also would like to point out that there exists considerable precedent in the datasets literature for publishing datasets consisting of links and metadata. Examples include the CC12M dataset (CVPR 2021), the CC3M dataset (ACL 2018), and LAION 400M (NeurIPS Workshop 2021). BioCLIP (CVPR Best Paper Award 2024) also released their image dataset only in part and does not release their metadata, requiring substantial work to fully recreate on a large-scale compute cluster. Our approach aligns with these practices, balancing accessibility and efficiency.
>
> In addition, we have also released a comprehensive tooling pipeline that helps in curating an AI-ready dataset; see our [git repo](https://github.com/baskargroup/Arboretum/tree/main/Arbor-preprocess/arbor_process). As far as we are aware, there are no free/open tools which provide these capabilities.
>
> **Regarding the suggestion to provide additional benchmarks on other architectures:** we fully acknowledge the importance of adding more benchmark results. Unlike prior work such as BioCLIP (CVPR Best Paper 2024), our study does explore the choice of pretraining architecture by fine-tuning ArborCLIP checkpoints from both ViT-L-14 and ViT-B-16 bases. While further ablations could provide more context, the effect of architecture using similar datasets have been well-documented in the literature.
>
> That said, we are currently training ArborCLIP checkpoints for use in resource-constrained settings and plan to report these results as part of our camera-ready draft. We have also expanded our comparisons suite, adding benchmarks against iNaturalist 2017, iNaturalist 2019, GBIF Backbone Taxonomy, and Wildlife Insights. These additions are reflected in Table 2 and further detailed in a separate PDF (attached).
>
> **Regarding the dataset name, "Arboretum":** this name served several purposes --- it reflected our team’s interest (and expertise) in agricultural applications; harkened to the symbolic "tree of life",  representing the interconnectedness of all living organisms; and conveys the essential element of curation (which is central to our work). We are pleased to report that the name "Arboretum" has already gained some modest visibility in AI-for-agriculture circles and therefore we respectfully intend to retain it.
>
> **Regarding the iNaturalist dataset and the potential biases inherent in citizen science data:** The iNaturalist dataset benefits from the contributions of a large and engaged community of naturalists, biologists, and citizen scientists. Observations reach research-grade status when two or more members of the iNaturalist community, which includes experienced naturalists, biologists, and citizen scientists, agree on the identification, and the observation meets specific criteria such as having a photograph and accurate location data. This community-driven process ensures a high level of accuracy and reliability by leveraging the collective expertise of a large and engaged community, which includes global partner organizations such as the New Zealand Biodiversity Recording Network, Canadian Wildlife Federation, Instituto Humboldt, and many others.
>
> Recent experiments have demonstrated that the accuracy of iNaturalist Research Grade observations is approximately 97%, supported by this robust community validation process. Although biases may certainly exist that reflect the visibility of certain species and the effects of urbanization, we do not significantly detract from the dataset's overall value. The continuous updating of the iNaturalist dataset further ensures its relevance and reliability as a resource for biodiversity research.
>
> Again, we thank the reviewer for their thoughtful comments. We are happy to engage in further discussion and address any new questions that the reviewer may have regarding our work.
>
> **References:**
>
> 1. [iNaturalist Observation Accuracy Experiment](https://www.inaturalist.org/blog/90263-a-second-experiment-to-learn-about-the-accuracy-of-inaturalist-observations)
> 2. [Image Resolution Reference](https://forum.inaturalist.org/t/image-upload-size-limit-warning-should-be-dropped-as-images-are-automatically-resized-anyway/39645/18)
> 3. [Image Cropping Reference](https://www.inaturalist.org/pages/responses#crop)
> 4. [Piccolo, Renee Louise, et al. "Location biases in ecological research on Australian terrestrial reptiles." *Scientific Reports* 10.1 (2020): 9691.](https://www.nature.com/articles/s41598-020-66610-1)
> 5. [iNaturalist Open Data](https://github.com/inaturalist/inaturalist-open-data)

---

### Official Review · Reviewer_pSBJ · 2024-07-25
**Well organized curated dataset, with thorough experiments and discussion**

**Rating:** 8
**Confidence:** 4

**Review:**

The quality of work in this paper is high. The writing is very clear and is easily understood. The only issue found that is bothersome relates to the vetting of the data (see cons section).

Pros:
- Very well organized and clear
- Comparisons to existing work are made
- Design decisions are well-motivated (e.g., the inclusion of both common and scientific names)
- Challenges associated with the iNaturalist data source are explained
- Steps are taken to ensure only data with appropriate licenses were used
- A data preparation pipeline is provided and allows users to "select specific categories, visualize data distributions, and manage class imbalance effectively based on their needs. It facilitates the downloading of specific images by their URLs and provides image-text pairs and user-defined chunks to support various AI applications." This itself is a meaningful contribution, since it makes the data dramatically more usable for a variety of applications.
- Experiments are well designed
- Limitations of the work are described in an open manner (for example, the mention of poor performance on the confounding-species benchmark).
- Well presented project webpage, github, huggingface available, pypi package available. All very well done!

Cons:
- It is stated that the metadata (labels) are vetted by domain experts; however, this claim is not substantiated nor elaborated upon. While as a user I am inclined to trust the claims of this paper, as a reviewer I must remain skeptical. Who are these experts? Vetting a dataset of this size is an incredible feat and would likely cost quite substantial amounts for paid experts. No mention of a grant or funding source was given (unless I missed it).
- To access the dataset, a sign-in to HuggingFace is required. This is not a serious issue (and may actually be reasonable). I did not access the data (although it does appear to be readily available on HuggingFace) because it requires a sign-in (and in particular, I would like to remain fully anonymous for this review).

**Strengths:**

- Similar to 'pros' listed in my main review of this paper
- This paper presents both a dataset and benchmark, and certainly does push the envelope in the field of biodiversity datasets and benchmarks.
- The work is well done in all aspects (again, my only concern is the details of metadata vetting). There is a good project page and the dataset seems to be readily available. The code is nicely setup in a way that makes using this dataset easy, thereby promoting its use.

**Additional Feedback:**

None.

**Clarity:**

Paper is very well written and easy to read. Figures, diagrams, tables, webpages, etc. are also well done.

**Correctness:**

Everything looks great except for the claim surrounding metadata vetting. More details must be provided there.

**Documentation:**

Little detail is given on the dataset maintenance plan. A URL to the project page placed on the first page of this paper would be a nice addition.

**Ethics:**

No concerns.

**Limitations:**

Limitations were discussed. No negative societal impacts were mentioned, though I don't see anything obvious from my own assessment.

**Opportunities For Improvement:**

More explanation on the vetting process is required. The dataset, code, pipeline, project pages, etc. are all good, but the claims regarding quality and vetting of the dataset need more elaboration.

**Relation To Prior Work:**

Good comparisons to existing datasets and benchmarks are made.

**Summary And Contributions:**

This paper introduces a curated dataset based on the publicly available crowd-source / citizen-science data from iNaturalist. The dataset is the largest publicly available dataset of images for assessing biodiversity (at least in terms of number of images, rather than number of distinct species, though it is not far behind TreeOfLife-10M). It is stated that the metadata (labels) are vetted by domain experts; however, this claim is not substantiated nor elaborated upon. Details (tables, plots, figures, and discussion) are given on the dataset and its curation process. Comparisons are made with other similar state-of-the-art datasets, and a variety of experiments are conducted, and in particular, a benchmark foundation model, ArborCLIP, is introduced. The paper concludes with some discussion of limitations and future work.

---

> ### Author Rebuttal · Authors · 2024-08-17
>
> Thank you for your thorough review. We appreciate your positive feedback on the overall quality, clarity, and organization of our work, and for recognizing the contributions our work makes to the field of biodiversity datasets and benchmarks.
>
> Below, we address your specific concerns:
>
> **Regarding metadata vetting:** We understand your concern regarding the vetting of the metadata. The metadata in our dataset is based on research-grade observations from the iNaturalist platform. Observations achieve this status when they meet specific criteria, such as having a photograph and accurate location data, and when two or more members of the iNaturalist community—comprising naturalists, biologists, and citizen scientists—agree on the identification. This community-driven process leverages the collective expertise of a large, diverse community, ensuring accuracy and reliability. iNaturalist's global partners, such as the New Zealand Biodiversity Recording Network and the Canadian Wildlife Federation, contribute to maintaining the quality of these observations. We will expand on this process in the revised manuscript for clarity.
>
> **Regarding dataset maintenance:** We will include a section in the revised manuscript outlining our plan for regularly updating and maintaining the dataset. Additionally, we will add a URL to the project page on the first page of the paper for easier access to the latest information and tools, including any updates or errata.
>
> **Regarding HuggingFace access:** We understand the need for anonymity during the review process. The dataset is hosted on HuggingFace, where sign-in is required primarily for tracking community engagement. The dataset is well-documented and accessible, and we invite you to explore it after the review process. We are happy to address any further questions at that time.

---

> > ### Comment · Reviewer_pSBJ · 2024-08-30
> >
> > Thank you for the clear response to my concerns.
> >
> > Regarding data vetting: Thank you for clarifying this process. It would indeed be good to elaborate on this within the manuscript. However, I still feel that the claims regarding data vetting should be softened. For example, simply having two citizen scientists agree upon the identification does not constitute vetting by "domain experts" in my opinion (in other words, I would be mislead by this claim). I realize that you will provide further details, but I do still feel that that a more modest claim would be more accurate and less misleading.
> >
> > I am happy to keep my rating as-is and hope that you do consider my additional feedback given here.

---

### Official Review · Reviewer_cn6Q · 2024-07-30

**Rating:** 6
**Confidence:** 3
**Clarity:** Generally, yes.

**Review:**

A generally well-constructed dataset, which greatly expands the size of the latest biodiversity datasets. The paper uses an interesting evaluation paradigm, though the results could have been presented more clearly and it would have been nice to have further evaluation (e.g. genera vs species results; few-shot classification accuracy) if not more experiments.

**Strengths:**

I like the evaluation on a variety of datasets (balanced, unseen, and life stages).

The curation and partitioning appears to have been done with care.

**Additional Feedback:**

L195 Please specify dates in ISO standard YYYY-MM-DD or write out the month in full (as done on L176) to prevent confusion for an international (non-American) audience.

Section numbering in the appendix is not correct. Currently Section 7 is named Appendix, but then Section 8 and 9 follow it. Are these not part of the appendix? The sections of the appendix should follow an `\appendix` command, which renders section headings as alpha instead of numeric mode, to make them distinct from the main text numbering. Section 7.6 is empty. Is this supposed to contain the composition of the 134.6M dataset?

**Figures**

Fig 3: Great to include a treemap to show the data. However, it might be easier to read if the shading of the boxes changed by nesting level? At the moment, it is hard to see where the class and order boxes span as they are so large (e.g. Magnoliopsida).

Fig 4b: I think it would be more intuitive to have the stages progress chronologically from left-to-right, with each row being a species. So I recommend flipping the orientation of the figure.

Fig 5: It is hard to read figures accurately when there are no ticks. e.g. what is the extent of the first bar in the histogram? Please add x and y ticks to these figures.


**Tables**

Table captions are in italic, which is undesirable as it is less easy to read a large block of italic text (e.g. Table 5 caption).

Numeric columns shouldbe right-aligned, with consistent formats within each table, for readability.

Thousands separators should be used consistently across tables (currently some use commas, some use siunitx spaces, others don't have any).

Table 3: "We integrate taxonomic labels into the images." - not clear what is meant by this. There are two sub-tables, and the caption should ideally reference them both explicitly e.g. Training data sources (left) and Diversity in Different Taxonoy (right). Alternatively, give each its own caption (i.e. convert to Table 3 and Table 4).

Table 4: Examples column heading misaligned.

Table 5:
- Caption refers to "ArborCLIP-O", "-B", "-M", that are not the names used in the table.
- Adding a citation to the datasets used in Table 5 that are existing benchmarks (e.g. Birds525) would make it clearer which are pre-existing and which are proposed by the paper.
- Some numbers are given with 1dp, and others 0dp, and numbers are center-aligned; this all makes it hard to read the values in the table and compare them against each other at a glance.
- The highlighting is not described in the caption. It seems to be best in purple and second best in blue, in which case I think there should be two purple and zero blue highlights in column AB.
- Some columns test on classes that were seen during training, and others (AU) classes that were not seen during training. Perhaps this could be indicated with an extra heading level in the table?
- It would be helpful to indicate which evaluations were done with scientific names and which with common names in the table.

Table 6: Should be a booktabs table like the rest. Category should be left-aligned. It may be beneficial to have another column for the rank named instead of integrating the rank name and value into one column.


**Typographical errors**

- Fig 1 caption: Careful with opening quotemarks (use a backtick in LaTeX source).
- L142 "interesitng"
- L219 450K -> 450k
- Section 6 should be title "Conclusion" or "Discussion", not "Concluding Discussion"
- Many citations are cased incorrectly: [12], [15], [20], [24], [30], [32], [34], [37], [38], [41].
- Some citations lack complete reference information: [4], [18], [41]
- Citation [8] has arXiv formatted differently from other arXiv citations.

**Correctness:**

It is not clear whether the 5% test batches are sufficiently stratified to contain at least one sample from each species. Can the authors please confirm this?

**Documentation:**

I believe there are sufficient details.

**Ethics:**

No. The authors say they have removed personal details from the dataset.

**Limitations:**

I think this was adequately addressed.

**Opportunities For Improvement:**

For the title, I don't think it is suitable as the modalities are only images and label-like captions. The text modality is not free-form it is highly constrained.

The dataset is very big... but some might say too big! Do we really need half a million images of species Anas platyrhynchos? Considering that the experiments are all performed on the Arboretum-40M dataset, which has a sensible limit of 50k samples per species. Perhaps that should be the final dataset, instead of the rather bloated 134.6M image full dataset?

It is not clear that the filtering described in Section 3 applies to the Arboretum-40M subset only (described in Sec 4.1), and not the whole Arboretum dataset. Sec 3 also doesn't provide the max limit of 50k value, which is only in Sec 4.1. It would be better to re-arrange this content so Arboretum-40M is described in one place.

For Arboretum-Unseen, evaluation is performed on species that were not seen during ArboretumCLIP training. Do the authors know whether these species were seen during the training of the pretrained models (OpenAI-CLIP, BioCLIP, MetaCLIP)?

The inconsistency in the number of training epochs For ArborCLIP-O, -B, -M (40, 8, 12) is not justified in the paper. What was the motivation for this?

The evaluation protocol is not described in the main text, which it should be.

The work would benefit from multi-shot evaluation to accompany the zero-shot evaluation.

It would be helpful to have stats for the Arboretum evaluation datasets (number of samples, classes, class distribution) shown in a table together. Perhaps integrated with Table 4. Is Table 4 the stats for datasets per evaluation? It is unclear which split is shown and which split is used?

For Arboretum-Unseen, it would be helpful to see the performance when using scientific names down to genus level, as the model has (presumably) seen each genera in the training data. I would like to know the model's ability to operate on the species-level names beyond what it can manage at the genus-level.

The authors don't include any statistical analysis, but they have multiple benchmark datasets and take the average performance over those. Might I suggest they do statistics over the evaluation datasets, in lieu of random seeds? Such a test can be done using the Wilcoxon signed-rank test between a pair of models using their results for each evaluation metric.

**Relation To Prior Work:**

Yes.

**Summary And Contributions:**

The paper provides a new very large dataset of 134.6M labelled images taken from the iNaturalist Open Dataset. A model is trained on a 40M subset of it in the style of CLIP, and evaluated on several existing datasets and partitions that the authors propose.

---

> ### Author Rebuttal · Authors · 2024-08-17
>
> Thank you for your careful review and valuable feedback.
>
> **Regarding the text data:** We acknowledge that the associate text follows a structured format. This was an intentional choice to ensure consistency across the dataset. Text entries, such as "a photo of Convergent Lady Beetle" or "a photo of kingdom Animalia phylum Arthropoda class Insecta order Coleoptera family Coccinellidae genus Hippodamia species convergence," are designed to provide clear, precise information that aligns with the corresponding images.
>
> While this format may appear constrained, it serves a specific purpose: to ensure that the model can reliably associate images with their correct labels and taxonomic classifications. This consistency is important for training models in tasks that require precise recognition and classification of biological entities. We appreciate your suggestion and agree that exploring less constrained text formats could be valuable; in future releases of the dataset, we will include other text forms.
>
> **Regarding the concern about the number of samples:** This dataset was derived from iNaturalist Open Data, where *Anas platyrhynchos* is one of the most frequently observed species. This size is not set in stone, and users have the flexibility to reduce the number of images per species to match their specific research needs. For example, we curated the Arboretum-40M dataset as a testbed for our model training experiments, offering an even more balanced subset with a 50k sample cap per species. This is not the final or only version of the dataset. Future research may feature expanded subsets with 80M or 120M images, depending on specific study objectives.
>
> We appreciate your feedback and recognize the need for greater clarity regarding the filtering process. The data filtering and preprocessing tools (Section 3) apply to the entire Arboretum dataset. The Arboretum-40M subset was deliberate and not solely based on the 50k per species cap. We also set an upload date cutoff of January 27, 2024, for images uploaded to iNaturalist, with the 50k cap applied as part of this process. We agree that consolidating the discussion of the Arboretum-40M subset into one section would enhance clarity, and we will revise the manuscript accordingly.
>
> **Regarding species overlap:** Arboretum is sourced from iNaturalist. We confirm that BioCLIP utilized iNat-21 data, so there is likely some overlap between species in Arboretum-unseen (the benchmark dataset described in Section 4.2) and those seen during the training of BioCLIP. Regarding OpenAI-CLIP and MetaCLIP, these models were trained on large datasets of internet-sourced images, which may include iNaturalist data, but we do not have details about either pre-training dataset.
>
> **Regarding training:** As our ArborCLIP models were expensive to train on a per-epoch basis, we implemented early stopping for -B and -M based on the average contrastive loss per epoch, with a patience of one epoch; this accounts for the inconsistency in the stopping epoch.
>
>
> **Regarding the evaluation protocol:** We appreciate the feedback. While we discussed aspects of the evaluation process in the paper, we acknowledge that it was not clearly highlighted in the main text. We will revise the manuscript accordingly. We agree that deploying multi-shot classification methods such as SimpleShot can greatly improve the performance of CLIP-style classifiers on long-tail classes, as can other reasonable approaches like Nico++, REAL, Tip-Adapter, and others. Instead of delving deeply into multi-shot algorithms, we chose to focus on a simple evaluation protocol (zero-shot classification) that better reflects the use of our models in practice.
>
> We recognize the importance of providing clear statistics for the Arboretum evaluation datasets, including the number of samples, classes, and class distribution. Detailed statistics for the Arboretum-40M subset are provided in the appendix; the benchmarks and their relevant statistics are in the main text. To ensure the integrity of the evaluation process, we took specific measures to avoid overlap between the training and evaluation datasets. Specifically, 5% of the mini-batches were reserved exclusively for testing, separate from the 95% used for training and validation, with random selection applied after our semi-global shuffling strategy. Additionally, benchmark datasets like Arboretum-Unseen were curated by excluding species with fewer than 30 instances in the main Arboretum dataset, ensuring no overlap with the training data. Likewise, the Arboretum-Lifestages dataset was collected from February 1, 2024, to May 20, 2024, after the training dataset was finalized, using life stage filters to prevent overlap. We will revise the manuscript to integrate these details more effectively, consolidating them into Table 4. This will ensure that all relevant information is presented together.
>
> We agree that adding statistical analysis would strengthen our findings. We have incorporated this suggestion in the rebuttal; please see the 95% binomial proportion confidence intervals to the scores we report in Table 5. Though the 5% test batches were originally selected after a random shuffle of the dataset, we can improve this approach by first obtaining the sample distribution for each species and then randomly selecting 5% based on this class distribution. This method could help ensure that the test batches contain samples from each species, providing a more representative evaluation.
>
> There is considerable precedent in the literature for publishing datasets consisting of links and metadata, including the CC12M dataset (CVPR 2021), the CC3M dataset (ACL 2018), and LAION 400M (NeurIPS Workshop 2021). BioCLIP (CVPR Best Paper 2024) releases its image dataset only in part and does not release its metadata, requiring substantial work to recreate it on a large-scale compute cluster. Our approach aligns with these practices, balancing accessibility and efficiency.

---

> > ### Comment · Reviewer_cn6Q · 2024-08-22
> >
> > Thank you to the authors for their response. I will take it into consideration.
> >
> > Can the you clarify for me, what you have added to the dataset that wasn't in iNaturalist already?
> >
> > > We confirm that BioCLIP utilized iNat-21 data, so there is likely some overlap between species in Arboretum-unseen (the benchmark dataset described in Section 4.2) and those seen during the training of BioCLIP.
> >
> > Can you investigate this further and quantify the overlap? This seems highly relevant when BioCLIP and ArborCLIP from BioCLIP are the best performing models, and this may be explained by the model having already seen species which are supposed to be held-out from test during the BioCLIP training stage.
> >
> > > Though the 5% test batches were originally selected after a random shuffle of the dataset, we **can** improve this approach by first obtaining the sample distribution for each species and then randomly selecting 5% based on this class distribution. This method could help ensure that the test batches contain samples from each species, providing a more representative evaluation.
> >
> > But will you?
> >
> > > we implemented early stopping for -B and -M based on the average contrastive loss per epoch, with a patience of one epoch; this accounts for the inconsistency in the stopping epoch.
> >
> > Will you add this to the manuscript?
> >
> > > There is considerable precedent in the literature for publishing datasets consisting of links and metadata...
> >
> > I think you have me mistaken for somebody else, as I didn't criticize this.

---

### Author Rebuttal · Authors · 2024-08-17

We are grateful for the thorough and constructive feedback provided by the reviewers. Your comments have been invaluable in refining our work, and we are committed to addressing them to further strengthen our submission.

**Dataset name**
The choice of the name "Arboretum" for our dataset was carefully considered to reflect our dual focus on agriculture and the symbolic "tree of life," which represents the interconnectedness of all living organisms and the cycle of life. The name "Arboretum" for the dataset is beginning to gain some visibility within the community, and therefore we would like to retain it. However, we understand the concern and are open to further discussion about this.

**Dataset hosting**
While Arboretum sources images from iNaturalist open data, our contribution goes far beyond merely providing metadata. We have significantly enhanced the dataset by adding ten new text descriptions for each image, which greatly improves its usability for AI and machine learning applications. This addition ensures that Arboretum is a rich, context-specific dataset that is both new and unique.

Given Arboretum's scale of over 130 million images, independently hosting it would incur substantial financial and environmental costs. Since the iNaturalist dataset is continuously updated, managing these updates separately could lead to redundancy and outdated data. Our approach aligns with established practices in the literature, where datasets consisting of links and metadata are commonly published, as seen in the CC12M dataset (CVPR 2021), CC3M dataset (ACL 2018), and LAION 400M (NeurIPS Workshop 2021). By providing the dataset through URLs, alongside a comprehensive pipeline that makes the data AI-ready, we allow researchers the flexibility to apply user-defined filters and processing steps, making the dataset more adaptable to various research needs.

**Quality of iNaturalist Labeling**
We emphasize the high quality of iNaturalist data in Arboretum, where observations achieve research-grade status when two or more iNaturalist community members—experienced naturalists, biologists, or citizen scientists—agree on the identification and the observation meets criteria like having a photograph and accurate location data. This community-driven process ensures high accuracy and reliability by leveraging the collective expertise of a large, engaged community.

The diverse iNaturalist community, supported by global partners like the New Zealand Biodiversity Recording Network and Canadian Wildlife Federation, plays a crucial role in ensuring observation quality and accuracy. Recent experiments have demonstrated that the accuracy of iNaturalist Research Grade observations is approximately 97%, supported by this robust community validation process. Additionally, iNaturalist has implemented several measures to enhance data quality, including refining validator criteria and introducing new data quality assessment conditions.

**Addressing Common Concerns**
Some concerns were raised by multiple reviewers, particularly regarding the structured nature of text modality and the need for more benchmarks.  The structured text format was intentionally chosen for consistency and clarity, crucial for training models in precise recognition and classification. This consistency is particularly important for training models in tasks that require precise recognition and classification of biological entities. However, we acknowledge the value of exploring less constrained text formats and agree that adding variety could enhance the dataset.

We’ve also included new benchmark performance results for a version of BioCLIP trained solely on iNat21. Detailed in the attached PDF, our analysis shows that models trained on Arboretum outperform those trained solely on iNat21 across various benchmarks, especially in categories like Arboretum Unseen, Fungi, and Insects-2. While results on benchmarks like Life-Stages and DeepWeeds show moderate differences, this comparison highlights Arboretum’s advantages in enhancing model accuracy and effectiveness.

We have also expanded our dataset comparisons, adding benchmarks against iNaturalist 2017, iNaturalist 2019, GBIF Backbone Taxonomy, and Wildlife Insights. These additions are reflected in Table 2 and further detailed in the attached PDF. Not all of these datasets are “AI-ready”, which further highlights the distinctiveness of Arboretum.

Additionally, we have provided a detailed **Performance Comparison Across Datasets and Architectures** (with ± indicating 95% confidence intervals) in the rebuttal for the first reviewer, as well as a comprehensive **Comparison of Arboretum with Existing Biodiversity Datasets** in the rebuttal for the third reviewer. These comparisons further underscore the robustness and value of the Arboretum dataset within the broader context of biodiversity research and AI applications.

**References:**
1. [SimpleShot (arXiv 2019)](https://arxiv.org/abs/1911.04623)
2. [Nico++ (arXiv 2023)](https://arxiv.org/abs/2310.00158)
3. [REAL (CVPR 2024)](https://openaccess.thecvf.com/content/CVPR2024/papers/Parashar_The_Neglected_Tails_in_Vision-Language_Models_CVPR_2024_paper.pdf)
4. [Tip-Adapter (ECCV 2022)](https://www.ecva.net/papers/eccv_2022/papers_ECCV/papers/136950487.pdf)
5. [iNaturalist Observation Accuracy Experiment](https://www.inaturalist.org/blog/90263-a-second-experiment-to-learn-about-the-accuracy-of-inaturalist-observations)
6. [Image Resolution Reference](https://forum.inaturalist.org/t/image-upload-size-limit-warning-should-be-dropped-as-images-are-automatically-resized-anyway/39645/18)
7. [Image Cropping Reference](https://www.inaturalist.org/pages/responses#crop)
8. [Piccolo, Renee Louise, et al. "Location biases in ecological research on Australian terrestrial reptiles." *Scientific Reports* 10.1 (2020): 9691.](https://www.nature.com/articles/s41598-020-66610-1)
9. [iNaturalist Open Data](https://github.com/inaturalist/inaturalist-open-data)

---

> ### Comment · Reviewer_cn6Q · 2024-08-17
>
> 1. *Meaning of multimodal.*
>
> Merriam-Webster
> > **multimodal**, *adjective*
> >
> > having or involving several modes, modalities, or maxima
>
> Cambridge Dictionary
> > **multimodal**, *adjective (also multi-modal)*
> >
> > involving several ways of operating or dealing with something:
>
> Collins Dictionary
> > **multimodal**, *adjective*
> >
> > 1. (of a statistical distribution) having several modes or maxima
> > 2. characterized by several modes of activity
> > 3.  involving or using several modes or methods
>
> If this dataset is multimodal then every labelled dataset ever curated is multimodal, and the word becomes entirely useless to the field.
>
> For example, ImageNet-1k classes (0, ..., 999) have synset codes class labels e.g. n01440764, which can trivially be mapped with a look-up table to their names e.g. "n01440764 = tench, Tinca tinca". So by the authors' logic, if I convert all the ImageNet-1k labels into text "a photo of <SYNSET NAME>", e.g. "a photo of tench, Tinca tinca", then it is now a multimodal dataset. This example string I show for ImageNet-1k is essentially indistinguishable from the "text modality" the authors use.
>
> Additionally, the iNaturalist datasets (e.g. iNaturalist-2021) do not claim to be multimodal, and this dataset is derived from exactly the same data sources as they are.
>
> 2. *Meaning of Arboretum.*
>
> Merriam-Webster
> > **arboretum**, *noun*
> >
> > a place where trees, shrubs, and herbaceous plants are cultivated for scientific and educational purposes
>
> Cambridge Dictionary
> > **arboretum**, *noun*
> >
> > a large garden where many types of tree are grown, for people to look at and to be studied for scientific purposes
>
> Collins Dictionary
> > **arboretum**, *countable noun*
> >
> > An arboretum is a specially designed garden of different types of trees.
>
> I have reviewed the paper a couple of weeks ago and I still think of it as "the tree dataset paper". Even though there are no trees, or in fact any plants, in the dataset.
>
> ---
>
> If the authors do not change the name of the dataset and paper, I will lower my score to recommend rejection. I am sorry to be blunt, but it is not my concern that the authors already circulated the preprint and dataset under the current name before it was reviewed.

---

> > ### Author Response · Authors · 2024-08-18
> >
> > Thanks for the quick response! We in fact appreciate the frankness and value such detailed and thoughtful feedback. We certainly hope to address each of your (and the other reviewers') points so that at the end of the discussion period you are comfortable recommending acceptance.
> >
> > We hear you on both points:
> > * *Multimodality* - this is a fair point, and your ImageNet-1K (counter)example is compelling. There is a bit more nuance: we use full taxonomy captions, and the taxonomic structure embedded within the serialized class label names enables a particular type of robust visual representation via vision-language training (this was indeed one of the key principles of BioCLIP by Stevens et al, 2024). But we agree this is not the way in which the AI community uses the term "multimodality". Therefore, we are happy to remove "multimodal" from both the title and the description of the dataset, and just refer to it throughout the paper as a curated dataset of images annotated with taxonomy and common name information.
> > * *"Arboretum" as the dataset name* - This is also a fair point, and the fact that this concern was echoed by two other reviewers makes it clear to us that a change might be necessary. We will attempt to converge to a better name over the next 2-3 days, and will post a global comment to this regard shortly after.
> >
> > Would these two modifications be acceptable? If there are other ways in which we can improve the paper, we are happy to address them through the rest of the discussion period.

---

> > > ### Comment · Reviewer_cn6Q · 2024-08-20
> > >
> > > > this was indeed one of the key principles of BioCLIP by Stevens et al, 2024
> > >
> > > Yes, indeed. Note that they do not actually claim their model is multimodal in "BioCLIP: A Vision Foundation Model for the Tree of Life" - they just claim it is a vision model. Not only is multimodality of the model not claimed in the title, but actually nowhere in the text either. They only use "multimodal" to refer to the CLIP model and the training objective.
> > >
> > > For the paper under review here, IMO it is important that "multimodal" is removed from the title, and succinct descriptions of the dataset. But the term can appear in the text of the paper if it is done appropriately, e.g. "this is a multimodal dataset" is bad because it implies the data has multiple sensory domains, whereas "we deploy multimodal CLIP training techniques" is fine, like is used in BioCLIP.
> > >
> > > > Would these two modifications be acceptable?
> > >
> > > Yes, that sounds great, thank you.

---

### Author Response · Authors · 2024-08-20
**Proposed change in dataset name, paper title**

Dear reviewers (and AC),

thanks very much for the productive discussion thus far. Based on the feedback from the reviewers, we would like to propose two top-level changes to the paper. While we do not see an obvious way to modify the manuscript or OpenReview metadata at this point, we will commit to implementing the following revisions:

* We will change the name of our dataset to **Biotrove**. This new name reflects our focus on biodiversity, and the suffix "-trove" reflects the value (we hope) the dataset will provide for the community. This is in response to feedback from 3 reviewers (cn6Q, KAB1, 7g25),
* We will drop the term "multimodal" from the paper title and any descriptions of the dataset within the text and the appendix. This is in response the feedback from reviewer cn6Q. We will simply call it a large, curated dataset of images of a diverse set of species.

Therefore, the new paper title will be: **Biotrove: A Large Curated Image Dataset Enabling AI for Biodiversity**.

We hope these changes will be acceptable to all. Thanks once again, and we welcome any further questions and/or comments.

---

### Decision · Program_Chairs · 2024-09-26

**Decision:**

Accept (Spotlight)

**Comment:**

This paper introduces Biotrove (formerly Arboretum) currently the largest curated dataset for biodiversity applications. Reviewers agree that this is a well-organized and clear contribution, ahead of the state of the art. It also contributes clear comparisons with concurrent work, on a a variety of tasks.
While concept of a dataset like this is not new, the scale of the dataset is a concrete step ahead what makes this contribution significant. Species identification is a very active topic in ML research, and this benchmark may enable future ML research.
Design choices are well motivated, and authors discussed some of the shortcomings with the reviews. I would appreciate some further motivation about the definition of the new benchmark tasks, esp compared to previous work, and how they are different from existing one.
I expect the depth of the discussion during the rebuttal to be reflected in the final manuscript.